# Forget Forgetting: Continual Learning in a World of Abundant Memory

**Dongkyu Cho[1], Taesup Moon[2], Rumi Chunara[1], Kyunghyun Cho[1], Sungmin Cha[1]***
[1]New York University   [2] ECE / ASRI / IMNC / IPAI, Seoul National University
*dongkyu.cho@nyu.edu, tsmoon@snu.ac.kr, {rumi.chunara, kyunghyun.cho, sungmin.cha}@nyu.edu*

## Abstract

Continual learning (CL) has traditionally focused on minimizing exemplar memory, a constraint often misaligned with modern systems where GPU time, not storage, is the primary bottleneck. This paper challenges this paradigm by investigating a more realistic regime: one where memory is abundant enough to mitigate forgetting, but full retraining from scratch remains prohibitively expensive. In this practical "middle ground", we find that the core challenge shifts from stability to plasticity, as models become biased toward prior tasks and struggle to learn new ones. Conversely, improved stability allows simple replay baselines to outperform the state-of-the-art methods at a fraction of the GPU cost. To address this newly surfaced trade-off, we propose Weight Space Consolidation, a lightweight method that combines (1) rank-based parameter resets to restore plasticity with (2) weight averaging to enhance stability. Validated on both class-incremental learning with image classifiers and continual instruction tuning with large language models, our approach outperforms strong baselines while matching the low computational cost of replay, offering a scalable alternative to expensive full-retraining. These findings challenge long-standing CL assumptions and establish a new, cost-efficient baseline for real-world CL systems where exemplar memory is no longer the limiting factor.

## 1 Introduction

As machine learning systems are increasingly considered to be deployed in dynamic, real-world environments, continual learning (CL) has emerged as a critical paradigm for adapting to evolving data streams without catastrophic forgetting (Wang et al., 2024a). A central challenge in this setting is the *stability–plasticity dilemma* (Carpenter & Grossberg, 1987; Mermillod et al., 2013): models that maintain high stability across prior tasks often fail to incorporate new knowledge (high stability, low plasticity), while those that remain highly plastic tend to forget earlier information (high plasticity, low stability). Various CL scenarios have been explored—most notably in class-incremental learning (class-IL) for image classification (Masana et al., 2022), and more recently in the continual learning of large language models (LLMs) (Wang et al., 2024a; 2023). Among CL approaches, *exemplar-based methods*—which store and replay representative samples from past tasks—have become particularly popular due to their simplicity and effectiveness (Masana et al., 2022; Zhou et al., 2024; Wang et al., 2023). However, a notable trend in these methods is the use of *highly constrained memory budgets*. For instance, many class-IL benchmarks assume only 20 exemplars per class—roughly 4% of the total training data—are retained across tasks (Rebuffi et al., 2017; Zhou et al., 2024). Similarly, LLM-focused CL approaches often rely on restricted caches or memory-free mechanisms to sidestep the issue of long-term storage (Wang et al., 2023). Yet the necessity and realism of such severe memory constraints remain questionable (Chavan et al., 2023; Yousuf Harun et al., 2023), and practical solutions to address this gap are still underexplored.

In real-world machine learning deployments, this assumption of severely limited exemplar memory is often misaligned with practical constraints (Prabhu et al., 2023). Modern storage solutions—such as cloud-based object stores or local SSDs—are both affordable and scalable. In contrast, GPU time, especially for training large-scale foundation models (*e.g.*, LLMs), constitutes a significant bottleneck. For example, an AWS instance with 8 A100 GPUs can cost over $30 per hour, while storing 1TB

---

*Corresponding author.

of data costs less than \$25 per month (Amazon Web Services, 2024). If the primary goal of CL is to enable efficient model adaptation to non-stationary data without full retraining from scratch (*i.e.*, avoiding expensive joint training), then *reducing GPU cost*—rather than storage usage—should be the main optimization objective.

Building on this observation, we revisit the CL setting under more realistic scenarios where exemplar memory constraints are relaxed. Our analysis reveals a critical trade-off across the memory spectrum: while traditional memory-constrained setups are often unrealistic and full-data retraining (joint training) is prohibitively expensive, a practical "middle ground" of abundant-but-not-exhaustive (*i.e.*, *sufficient*) memory regime emerges. It is precisely in this realistic regime that we identify a new challenge: while stability improves due to reduced forgetting, plasticity diminishes as the model becomes biased toward prior tasks. This highlights the urgent need for cost-efficient mechanisms that can restore plasticity without sacrificing stability, a gap our work aims to fill. We further investigate how this regime affects existing CL methods across two domains: class-IL and continual instruction tuning of LLMs. In class-IL , we observe that many state-of-the-art methods incur significantly higher GPU training costs yet offer marginal improvements over naive replay baselines. In continual instruction tuning, common model merging strategies often suffer from limited plasticity or require storing a separate model per task, which limits scalability.

These limitations across both domains point to the need for a new approach that is cost-efficient yet effective under abundant exemplar memory regimes. Motivated by this, we propose *Weight Space Consolidation*, a simple yet effective method that operates directly in the model's weight space. It combines (1) *ranking-based parameter resets*, which periodically reset the dormant parameters (measured via gradient-based signal accumulation) to their pretrained values to restore plasticity, and (2) *weight averaging*, which maintains a running average of model weights during training to encourage convergence toward flatter, more stable optima. By modifying weights, our approach facilitates fast convergence without additional compute overhead in GPUs. Across class-IL benchmarks and LLM continual instruction tuning, our method consistently matches or surpasses the performance of state-of-the-art methods while maintaining training costs comparable to naive replay. These results demonstrate that, when exemplar memory is no longer the bottleneck, cost-efficient CL is both achievable and practical. Our contributions are summarized as follows:

- We revisit continual learning under relaxed exemplar memory constraints and show that even naive replay can achieve strong performance while significantly lowering GPU costs.
- We conduct an extensive analysis across the memory spectrum to reveal the stability-plasticity trade-off, demonstrating that in the "practical" abundant memory regime, restoring lost plasticity becomes as critical as preserving stability.
- We propose a lightweight and practical method, Weight Space Consolidation, which combines ranking-based parameter resets and weight averaging to address the stability–plasticity trade-off.
- We validate our approach across class-IL tasks (*e.g.*, CIFAR-100, ImageNet-100) and LLM continual instruction tuning (TRACE; Wang et al. (2023)), demonstrating consistent accuracy improvements and 3–4$\times$ cost reductions over state-of-the-art methods.

## 2 RELATED WORKS

**Continual learning.** Continual learning (CL) has been actively studied in various scenarios and methodological categories. Among the three scenarios of CL (Van de Ven & Tolias, 2019), class-incremental learning (class-IL) has been considered the most challenging and has been the most actively studied scenario (Masana et al., 2022). Generally, CL algorithms (including class-IL) can be categorized into regularization-based approaches, which penalize changes to important parameters for past tasks (Kirkpatrick et al., 2017; Aljundi et al., 2018; Cha et al., 2020; Kang et al., 2022), rehearsal-based approaches, which store and replay exemplars from past tasks (Rebuffi et al., 2017; Cha et al., 2023), and expansion-based approaches, which expand the model's capacity to balance the trade-off between stability and plasticity (Yan et al., 2021; Wang et al., 2022). Additional approaches focus on addressing classifier bias toward recent tasks while using the exemplars (Wu et al., 2019; Zhao et al., 2020). While exemplar-based methods have demonstrated state-of-the-art performance, they typically rely on strict memory constraints, often limiting memory size to a small percentage of the dataset (Rebuffi et al., 2017; Zhou et al., 2024). Recent studies challenge the necessity of these strict memory constraints, highlighting that the computational cost of maintaining and processing memory—especially GPU usage—can far outweigh storage costs (Prabhu et al., 2023; Chavan et al., 2023; Harun et al., 2023). This shift in perspective opens the door to relaxing memory limits in order

to reduce training costs, which is the focus of our work. Lastly, another line of work solely focuses on the loss of plasticity in CL (Dohare et al., 2024), where parameter resetting is commonly suggested as a solution (Ash & Adams, 2020; Galashov et al., 2024; Wang et al., 2024b; Farias & Jozefiak, 2024). In contrast, we show how both stability and plasticity are issues under realistic CL scenarios.

**Weight space operations.** A growing body of work has explored directly manipulating model parameters in weight space across various domains, including domain generalization (Wortsman et al., 2022; Cho et al., 2025a), multi-task learning (Yu et al., 2024; Yang et al., 2023), and continual learning (Marouf et al., 2025; Marczak et al., 2025; Dziadzio et al., 2024). Most of these approaches operate as post hoc methods by merging the weights of pretrained models. For instance, TIES (Yadav et al., 2024) proposes a selective merging strategy to mitigate interference between different tasks, while Ilharco et al. (Ilharco et al., 2022) demonstrate that simple arithmetic on task-specific weight deltas can edit models without further training. Building on these ideas, recent studies have extended weight-space operations to continual learning. Kozal et al. (Kozal et al., 2024) apply weight averaging techniques in a CL setting, and Marczak et al. (Marczak et al., 2025) introduce a selective merging approach tailored for continual adaptation. However, such methods typically require storing multiple full model checkpoints during training, fail in accumulating various task knowledge (Dziadzio et al., 2024), and more critically, may violate the sequential constraints of CL. In contrast, our method operates in weight space *during* training (Izmailov et al., 2018; Jang et al., 2025), requiring neither multiple model copies nor post hoc merging. This enables cost-effective and online editing of the model's parameters while maintaining compatibility with the CL setup.

**Positioning.** We clarify the positioning of our work in Table 1. Several recent papers have sought cost-effective methods for CL (Prabhu et al., 2023; Harun et al., 2023; Chavan et al., 2023), but do not expand their study on varying exemplar memory scenarios (*e.g.*, constrained/ abundant/ full). On the other hand, we enumerate our experiments across scenarios, illuminating the effect of exemplar memory sizes on the model's stability and plasticity. While some

Table 1: Comparison of continual learning papers across different criteria.

| Paper | Constrained Memory | Abundant Memory | Full Memory | Constrained Computation | Loss of Plasticity | Catastrophic Forgetting |
|---|---|---|---|---|---|---|
| Wang et al. (2022) | ✓ | | | | | ✓ |
| Prabhu et al. (2023) | ✓ | ✓ | | ✓ | | ✓ |
| Harun et al. (2023) | ✓ | | | ✓ | | ✓ |
| Chavan et al. (2023) | ✓ | | | ✓ | | ✓ |
| Dohare et al. (2024) | | | ✓ | | ✓ | |
| Galashov et al. (2024) | ✓ | | | | ✓ | |
| Wang et al. (2024b) | ✓ | ✓ | | | ✓ | ✓ |
| Ours | ✓ | ✓ | ✓ | ✓ | ✓ | ✓ |

works have already studied the loss of plasticity, they focus on extreme settings (*i.e.*, full memory) where only plasticity is considered (Dohare et al., 2024), neglecting the issue of forgetting (Galashov et al., 2024) or failing to consider the computation costs (Wang et al., 2024b). In contrast, we focus on a realistic setting (*i.e.*, sufficient memory) where both plasticity and stability are considered.

## 3 MOTIVATION

**Notation.** We generally follow the setting of continual learning (CL) (Masana et al., 2022; Zhou et al., 2024). We consider a sequence of $T$ tasks, each associated with distribution $P_t$. Let $\mathcal{D}_t$ be the training dataset for task $t$, where $\mathcal{D}_t \sim P_t$. The tasks are presented in order $t = 1, \cdots, T$. The model $F$ (its parameters $\theta$) does not retain explicit access to previous task datasets $\mathcal{D}_j$ for $j < t$, except via an exemplar memory buffer $\mathcal{M}$ of a capacity of $K$. Thus, at the training step of task $t$, the model updates its parameters $\theta$ using the combined data $D_t \cup \mathcal{M}_{1:t-1}$ and the task-designated loss $\ell(D_t \cup \mathcal{M}_{1:t-1}; \theta)$, where $\mathcal{M}_{1:t-1}$ includes selected exemplars from earlier tasks.

### 3.1 DEFINING THE SUFFICIENT EXEMPLAR MEMORY REGIME

Most prior works assume a strictly limited exemplar memory budget, such that $K \ll \sum_{t=1}^{T} |\mathcal{D}_t|$. Under this constraint, the memory buffer $\mathcal{M}$ can retain only a small subset of examples from each past dataset $\{\mathcal{D}_1, \cdots, \mathcal{D}_{t-1}\}$. For instance, common class-IL benchmarks typically allocate only 20 exemplars per class, which corresponds to approximately 4% of the total training data (Rebuffi et al., 2017; Zhou et al., 2024). As a result, the buffer provides only a partial approximation of the true task distributions $\{P_1, \cdots, P_{t-1}\}$, leaving the model vulnerable to catastrophic forgetting.

By contrast, motivated by real-world scenarios where storage cost is relatively low but GPU cost is high, we re-examine CL in a practical regime with *sufficient* memory—enough to mitigate forgetting but where full retraining remains computationally expensive. Therefore, we pursue a practical "middle ground" of abundant-but-not-exhaustive exemplar memory. We define the memory buffer

$\mathcal{M}$ to be *sufficient* if it can retain enough samples to approximate the distribution of each previous task $P_j$ for $1 \leq j < t$. Formally, we assume the total memory budget $K$ satisfies $K \approx \kappa \sum_{j=1}^{t-1} |\mathcal{D}_j|$, where $\kappa \in (0,1]$ determines the fraction of past task data that can be stored. A larger value of $\kappa$ implies that $\mathcal{M}$ contains a more representative subset of earlier examples, though not necessarily the entire datasets. In Section 3.2, we investigate when the exemplar memory size becomes *sufficient* by experimenting over various memory settings. Under this sufficient exemplar memory setting, the mixture distribution $P_{past}$ of previously encountered tasks at the training step of task $t$ becomes:

$$P_{past}^{(t)} \approx \frac{\sum_{j=1}^{t-1} \pi_j P_j}{\sum_{j=1}^{t-1} \pi_j}, \tag{1}$$

where $\pi_j$ denotes the relative importance (*e.g.*, proportional to the number of stored samples or task frequency) of each past task, and $P_j$ represents the corresponding data distribution. In practice, we approximate $\pi_j$ with its empirical counterpart $\hat{\pi}_j$, which can be estimated from the samples stored in the buffer. During training on task $t$, the aggregated past distribution $P_{\text{past}}$ is combined with the current task distribution $P_t$ to form the hybrid training distribution:

$$P_{train}^{(t)} \approx \lambda P_t + (1 - \lambda) P_{past}^{(t)}, \tag{2}$$

where $\lambda \in [0,1]$ is a factor that balances the influence of the current and previously observed tasks.

In the next section, we analyze how this training distribution under sufficient exemplar memory influences the model's learning dynamics, leading to improved *stability* but degraded *plasticity*.

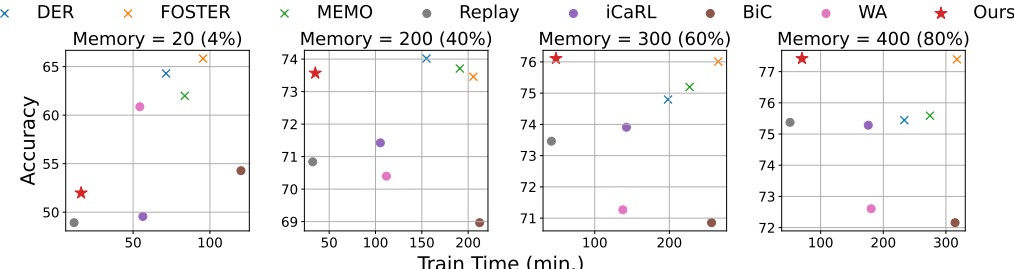

Figure 1: Comparison of (y-axis) average class-incremental accuracy and (x-axis) training time under different exemplar memory sizes in class-incremental learning for 10-task using CIFAR-100. As memory increases, catastrophic forgetting is mitigated (*i.e.*, increase in accuracy), but training time (*i.e.*, computation cost) also grows proportionally. Note that the DER, FOSTER, and MEMO are expansion-based methods (shown with X mark): FOSTER doubles the model size, while DER and MEMO scale with the number of tasks. Compared to these costly methods, Replay and Ours demonstrate high accuracy with significantly lower cost, where our method offers the highest cost efficiency, closely approaching that of the cost lower-bound cost (i.e., Replay).

### 3.2    WHAT CHANGES UNDER A SUFFICIENT EXEMPLAR MEMORY REGIME?

**Stability.**    Under sufficient exemplar memory, the distributions of previous tasks can be closely approximated, which effectively reduces forgetting—that is, improves *stability*. Specifically, with a large buffer $\mathcal{M}$, the empirical distributions $\tilde{P}_j$ of past tasks approximate their true distributions $P_j$ for $j < t$. This allows the empirical risk $\tilde{R}_{1:t}(\theta)$—computed over the stored exemplars—to closely approximate the ideal joint risk $R_{1:t}(\theta) = \sum_{j=1}^{t} \mathbb{E}_{x \sim P_j}[\ell(\theta; x)]$, as if the model were trained jointly on all tasks. As a result, the learned parameters $\tilde{\theta}_{1:t}^*$ remain close to the joint optimum $\theta_{1:t}^*$ in parameter space, preserving performance on previous tasks and mitigating catastrophic forgetting. For the complete derivation of this result, please refer to Appendix A.1.1.

Experimentally, Figure 1 shows that catastrophic forgetting is substantially reduced when exemplar memory is sufficient (*e.g.*, $|\mathcal{M}| \geq 200$). Notably, the simplest baseline (*i.e.*, Replay) outperforms more sophisticated methods while incurring significantly lower training cost (see Figure 1). As a result, by the end of each task $t-1$, the model serves both as a strong minimizer for previously learned tasks $1{:}t{-}1$ and as a reliable initialization for the upcoming task $t$.

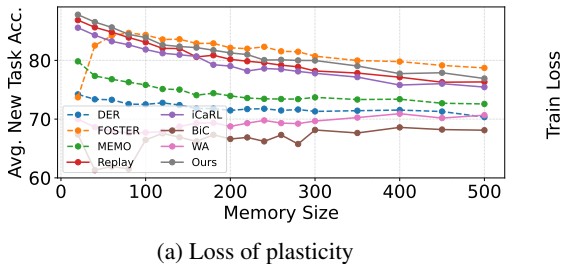 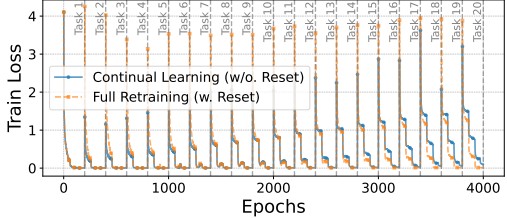

(a) Loss of plasticity

(b) Train Loss under full exemplar memory

Figure 2: Comparison of (a) average new-task accuracy under different exemplar memory sizes and (b) training loss under full memory in class-incremental learning for 10 tasks using CIFAR-100. As memory increases, the model's ability to adapt to new tasks declines, resulting in reduced accuracy and slower convergence. Notably, in (b), resetting model weights before each task restores plasticity and facilitates training.

**Plasticity.** We find that under this condition, the challenge shifts from stability to plasticity. We conjecture that as exemplar memory becomes increasingly sufficient, the model's ability to learn new tasks (*i.e.*, plasticity) gradually deteriorates. At task step $t$, the model is trained on a hybrid distribution $P_{\text{train}}^{(t)} \approx \lambda P_t + (1-\lambda)P_{\text{past}}^{(t)}$ (Eq. 2), where $\lambda \in [0, 1]$ controls the emphasis on the new task. As memory size increases, $\lambda$ decreases, causing the past-task distribution $P_{\text{past}}^{(t)}$ to dominate. When memory is sufficiently large, $P_{\text{past}}^{(t)}$ closely resembles $P_{\text{train}}^{(t-1)}$, and thus $P_{\text{train}}^{(t)} \approx P_{\text{train}}^{(t-1)}$. This similarity results in high gradient alignment (Du et al., 2018), which we quantify via cosine similarity $\rho_t = \langle \bar{g}_t^{(\text{new})}, \bar{g}_t^{(\text{past})} \rangle / \|\bar{g}_t^{(\text{new})}\| \|\bar{g}_t^{(\text{past})}\|$ (in Eq. 11), where $\bar{g}_t^{(\text{new})}$ and $\bar{g}_t^{(\text{past})}$ denote the mean gradients from the new tasks and past tasks, respectively. When $\rho_t \approx 1$, the expected gradient $\bar{g}_t = \lambda \bar{g}_t^{(\text{new})} + (1 - \lambda)\bar{g}_t^{(\text{past})}$ has a reduced norm, causing smaller updates (Yu et al., 2020):

$$\|\theta^{(t)} - \theta^{(t-1)}\| = \eta \|\bar{g}_t\| \leq \eta \|\bar{g}_t^{(\text{new})}\|, \tag{3}$$

which shrinks further as $\lambda \to 0$, where $\eta$ is the step size. This limits the model's capacity to adapt, as it tends to reuse previously learned parameters rather than learning new representations.

Such "parameter reuse" behavior has been observed when two sequential tasks are similar, leading to minimal drift across tasks (Lee et al., 2022; Dohare et al., 2024), which is a hallmark of low plasticity. Our interpretation aligns with the stability–plasticity dilemma (Mermillod et al., 2013; Zhang et al., 2024), where retaining prior knowledge comes at the cost of adapting to new information. It also corroborates observations from Rolnick et al. (2019), which showed that excessive exemplar memory can hinder the learning of new tasks. Please refer to Appendix A.1.2 for a more detailed discussion.

Figure 2a compares the model's average accuracy on new tasks across various CL methods, which is commonly used to measure the model's ability to acquire new knowledge (i.e., plasticity) (Liang & Li, 2023; Wang et al., 2024b). Here, we experimentally confirm that as memory increases, the model's plasticity generally degrades, resulting in lower average accuracy on new tasks (also see Table 6). This aligns with Figure 2b, which depicts the model's train loss under full exemplar memory. We can observe here that the large memory interferes with convergence under the CL setting. Notably in Figure 2b, we find that reinitializing the model parameters before each task (Dohare et al., 2024; Farias & Jozefiak, 2024) is a simple solution to restoring plasticity, as demonstrated by the low train loss of the model trained with reset and not continually (Orange) in Figure 2b.

## 4 WEIGHT SPACE CONSOLIDATION

Based on insights from the previous section, we propose a cost-efficient CL method—*Weight Space Consolidation*—that leverages weight space operations (*e.g.*, selective resets and running averages) to reconcile the trade-offs of the sufficient exemplar memory regime: high stability and low plasticity.

### 4.1 RANKING-BASED PARAMETER RESET FOR IMPROVED PLASTICITY

The core idea behind our method is that sufficient exemplar memory enables the model to start from a stable initialization (as discussed in Section 3), but remaining too close to this point can hinder plasticity. To address this, we introduce a ranking-based reset technique that selectively reinitializes less important parameters based on their estimated contribution to learning.

Before training on the $t^{\text{th}}$ task, the model parameters—optimized on previous tasks $1{:}t{-}1$—are stored as $\theta_{\text{prev}}$ (Line 6 in Algorithm 1). Training on the $t^{\text{th}}$ task proceeds using the task loss $\ell$ (Line 9), and after $n_{\text{warm}}$ warm-up epochs, we identify *dormant* parameters and gently reset them (Lines 10–11).

To rank parameter importance, we compute a moment-based metric $\mathcal{S}_l$ for each parameter element $l$, using the first and second exponential moments of its stochastic gradients $(\hat{m}_l, \hat{v}_l)$, as in Adam (Kingma & Ba, 2017):

$$\mathcal{S}_l = |\hat{m}_l| \cdot \hat{v}_l. \tag{4}$$

This score favors parameters that consistently receive strong gradients, while penalizing those with low or noisy updates. Specifically, a large $\hat{m}_l$ indicates that the parameter has received gradients in a consistent direction, while a large $\hat{v}_l$ implies that the parameter has experienced high gradient energy overall. By taking the product of the two, $\mathcal{S}_l$ becomes sensitive to both focused and sustained learning signals. Conversely, a low $\mathcal{S}_l$ suggests that the parameter has received weak or noisy gradients, indicating limited contribution to learning. We treat such parameters as *dormant* and reinitialize them to recover plasticity. This formulation provides a simple yet effective heuristic for identifying underutilized parts of the network based on gradient dynamics during warm-up. In Section 5, we compare existing methods that use weight resets for plasticity recovery (see Table 5).

In implementation, we retain only the top-$Q\%$ of the parameter element $l$ (with $Q{=}20$ by default following (Yadav et al., 2024)), and reset the rest by softly blending them with $\theta_{\text{prev}}$ (Lines 12–13):

$$\theta[l] = \alpha \cdot \theta[l] + (1 - \alpha) \cdot \theta_{\text{prev}}[l], \qquad \alpha = 0.5. \tag{5}$$

This gently pushes the model out of the previous solution basin to improve plasticity, while preserving parameters critical to prior tasks—thus maintaining stability.

Notably, this operation resembles model merging in the weight space, where two sets of parameters are blended. However, unlike conventional model merging approaches that combine multiple trained models post hoc, our reset mechanism is applied *during training*, with the explicit goal of restoring plasticity. In this context, we treat the merged weights not as a final model, but as an improved initialization point that facilitates adaptation to the new task without sacrificing stability.

## 4.2 WEIGHT AVERAGING FOR IMPROVED STABILITY

Using the reset model as a fresh starting point, we resume training $\theta$ for the remaining epochs. From this point on, we accumulate a running weight average $\Theta$ (see Lines 14–16) following the stochastic weight averaging (SWA) (Izmailov et al., 2018), which is known to promote convergence to flatter optima. The running average is updated every $j$ iterations after a warm-up phase of $n_{\text{warm}}$ steps:

$$\Theta \;\leftarrow\; \frac{n_{\text{avg}} \cdot \Theta + \theta}{n_{\text{avg}} + 1}, \tag{6}$$

where $n_{\text{avg}} = {i}/{j}$ and $i$ is the current iteration index.

We find that this averaging is particularly effective under sufficient exemplar memory settings, where data diversity introduces significant gradient variance. After the warm-up phase, the model often oscillates around multiple distinct low-loss regions due to this variance. By averaging weights across these regions, $\Theta$ converges to a flatter and more robust minimum that consolidates knowledge across both past and current tasks (Izmailov et al., 2018; Cha et al., 2020).

Importantly, our approach differs from traditional model merging methods in CL, which often combine independently trained task-specific models to construct a final model (Ilharco et al., 2022). In contrast, our method performs *in-situ* averaging during the training of a single model $\theta$, progressively updating

---

**Algorithm 1:** Weight Space Consolidation for cost-efficient CL

**1. Input:** Model parameters $\theta$, training data $D_{1:t}$, memory buffer $\mathcal{M}$, average interval $j$, warming epoch $n_{\text{warm}}$
**2. Output:** Trained model parameters $\theta$
**3. for** $t \leftarrow 1$ *to* $T$ **do**
**4.**      $\Theta \leftarrow \theta$ // Init. Averaged Model
**5.**      **if** $t > 1$ **then**
**6.**          $\theta_{\text{prev}} \leftarrow \theta$
**7.**      **for** $i = 1 : n_{\text{iter}}$ **do**
**8.**          Sample minibatch $b$ from $\{D_t \cup \mathcal{M}_{1:t-1}\}$
**9.**          Update $\theta$ using $\ell(b; \theta)$ and SGD
**10.**          **if** $(t > 1$ *and* $i = n_{\text{warm}})$ **then**
**11.**              $\mathcal{I}_{\text{reset}} \leftarrow$ FindDormantParams$(\theta, \theta_{\text{prev}})$
**12.**              **for** $l \in \mathcal{I}_{\text{reset}}$ **do**
**13.**                  Reset weights using eq. (5)
**14.**          **if** $(t > 1$ *and* $i > n_{\text{warm}}$ *and* $i\%j{=}0)$ **then**
**15.**              $n_{\text{avg}} \leftarrow i/j$
**16.**              $\Theta \leftarrow (\Theta \cdot n_{\text{avg}} + \theta)/(n_{\text{avg}} + 1)$
**17.**      $\theta \leftarrow \Theta$

$\Theta$ as a byproduct of the optimization trajectory. This eliminates the need to store and merge multiple per-task models, improving our method's cost-efficiency and scalability to longer task sequences.

At the end of training on task $t$, we replace the model parameters with the averaged weights $\Theta$ (see Line 17), which then serve as a stable initialization for the next task, preserving knowledge while enabling further adaptation. Please refer to Appendix A.2 for further implementation details.

**Summary.** Our method combines two simple yet effective weight-space operations to balance the stability–plasticity trade-off in the sufficient exemplar memory regime: (1) ranking-based resets recover plasticity by reinitializing dormant parameters, and (2) weight averaging enhances stability by converging to flat, robust minima. Both are directly motivated by our analysis in Section 3 and introduce negligible overhead, requiring no storage of per-task models or additional backward passes.

## 5 EXPERIMENT

### 5.1 EXPERIMENTAL SETTINGS

We evaluate our method's performance and cost-efficiency under various exemplar memory settings.

**Class-IL benchmarks.** We use two standard class-incremental learning benchmarks (Masana et al., 2022) via the PyCIL framework (Zhou et al., 2023a): **CIFAR-100**, an image classification dataset with 100 classes split into 10 sequential tasks (10 classes each), and **ImageNet-100**, a 100-class subset of ImageNet also split into 10 tasks of 10 classes each. We compare our method to seven exemplar-based class-IL baselines: iCaRL (Rebuffi et al., 2017), BiC (Wu et al., 2019), WA (Zhao et al., 2020), DER (Yan et al., 2021), FOSTER (Wang et al., 2022), MEMO (Zhou et al., 2023b), and Replay, a naive baseline that finetunes using current data and stored exemplars.

**LLM continual instruction tuning.** We also evaluate on TRACE (Wang et al., 2023), a continual instruction tuning benchmark for LLMs across 8 domains. We compare our method against Replay and 4 model-merging baselines, following recent findings that such methods can be effectively applied to CL (Roth et al., 2024; Dziadzio et al., 2024): Model Soups (Wortsman et al., 2022), SLERP (Jang et al., 2024), MagMax (Marczak et al., 2025), and Task Arithmetic (Ilharco et al., 2022).

**Architectures and protocol.** For image classification, we use ResNet-32 (He et al., 2016) on CIFAR-100 and ResNet-18 (He et al., 2016) on ImageNet-100 (He et al., 2016). For LLM experiments, we use an instruction-tuned version of LLaMA-3.2 (Grattafiori et al., 2024). All class-IL results are averaged over five seeds and reported as average class-IL accuracy across all tasks after the final step (Masana et al., 2022). For TRACE, final scores are reported after completing all sequential tasks. We select hyperparameters following the realistic CL evaluation protocol proposed by (Cha & Cho, 2025). Further details on experimental settings are in Appendix A.5.

### 5.2 EXPERIMENTAL RESULTS

**Class-IL results.** We report class-incremental learning results in Table 2 and Figures 1, 9. Note that DER, FOSTER, and MEMO are expansion-based methods that increase model size over time during training. To ensure a fair comparison, we follow the evaluation protocol of (Cha & Cho, 2025) and reuse the best hyperparameters found on CIFAR-100 when evaluating on ImageNet-100. From these results, we make three key observations: First, as shown in Table 2, under the conventional constrained memory setting (*e.g.*, 4% memory), existing class-IL methods outperform Replay. However, as the memory size increases, the performance gap narrows substantially. With 20% memory, most methods perform similarly to Replay, suggesting diminishing returns of algorithmic complexity in the abundant exemplar memory regime. Second, when training cost (*i.e.*, training time) is taken into account, Figures 1 and 9 (see Appendix A.4) show that expansion-based methods become highly inefficient. For instance, while FOSTER maintains strong accuracy even under abundant exemplar memory, its training time is 4–5$\times$ higher than that of Replay. Third, our proposed method—Weight Space Consolidation—demonstrates both strong performance and high efficiency. Table 2 shows that it consistently outperforms existing baselines (except for expansion-based methods) under abundant exemplar memory (*i.e.*, over 20% memory). Meanwhile, note that its training cost remains comparable to that of Replay, as shown in Figures 1 and 9. Together, these results validate that our method effectively mitigates the plasticity–stability trade-off in class-IL using abundant exemplar memory with minimal GPU computation overhead.

Table 2: Average class-IL accuracy (%) on CIFAR-100 and ImageNet-100 with varying exemplar-memory sizes. We report experimental results with varying memory sizes, ranging from 20 exemplars per class (a common setting in class-IL) to 400/600 exemplars per class (storing 80% of the previous dataset in CIFAR-100 and nearly half in ImageNet-100). Bold highlights the best non-expansion method. We report the standard error across 5 runs.

| | Memory Size (the number of exemplars per class)$_{\text{(ratio of memory to full data)}}$ | | | | | | | |
| | CIFAR100 | | | | ImageNet100 | | | |
| Method | $20_{(4\%)}$ | $80_{(16\%)}$ | $200_{(40\%)}$ | $400_{(80\%)}$ | $20_{(1.5\%)}$ | $200_{(16\%)}$ | $400_{(30\%)}$ | $600_{(46\%)}$ |
|---|---|---|---|---|---|---|---|---|
| DER | $63.95_{\pm1.9}$ | $70.13_{\pm1.6}$ | $74.64_{\pm1.1}$ | $75.60_{\pm0.9}$ | $71.96_{\pm0.6}$ | $78.59_{\pm0.7}$ | $79.61_{\pm0.5}$ | $80.53_{\pm0.6}$ |
| FOSTER | $66.22_{\pm1.6}$ | $67.67_{\pm1.7}$ | $73.53_{\pm0.8}$ | $77.28_{\pm0.5}$ | $70.14_{\pm0.7}$ | $76.01_{\pm0.7}$ | $80.94_{\pm0.6}$ | $82.79_{\pm0.6}$ |
| MEMO | $61.99_{\pm1.0}$ | $70.58_{\pm1.0}$ | $73.71_{\pm0.7}$ | $75.59_{\pm0.5}$ | $66.35_{\pm0.4}$ | $77.89_{\pm0.4}$ | $80.05_{\pm0.2}$ | $81.11_{\pm0.4}$ |
| Replay | $48.63_{\pm1.1}$ | $63.78_{\pm1.2}$ | $71.60_{\pm0.9}$ | $75.71_{\pm0.7}$ | $50.52_{\pm0.4}$ | $73.79_{\pm0.5}$ | $78.59_{\pm0.4}$ | $81.08_{\pm0.3}$ |
| iCaRL | $49.95_{\pm1.3}$ | $64.81_{\pm1.1}$ | $72.69_{\pm0.8}$ | $75.49_{\pm0.5}$ | $50.32_{\pm0.9}$ | $73.57_{\pm0.8}$ | $78.45_{\pm0.6}$ | $80.87_{\pm0.5}$ |
| BiC | $53.65_{\pm0.9}$ | $64.74_{\pm0.6}$ | $69.15_{\pm0.7}$ | $72.50_{\pm0.7}$ | $59.31_{\pm0.7}$ | $74.14_{\pm0.8}$ | $77.51_{\pm0.7}$ | $79.29_{\pm0.6}$ |
| WA | $\mathbf{61.32}_{\pm1.8}$ | $66.19_{\pm1.6}$ | $71.42_{\pm1.2}$ | $73.83_{\pm1.4}$ | $\mathbf{61.44}_{\pm1.1}$ | $75.85_{\pm0.8}$ | $78.79_{\pm0.8}$ | $80.21_{\pm0.8}$ |
| Ours | $52.16_{\pm1.2}$ | $\mathbf{66.89}_{\pm0.9}$ | $\mathbf{74.49}_{\pm0.8}$ | $\mathbf{77.71}_{\pm0.8}$ | $54.97_{\pm0.5}$ | $\mathbf{76.43}_{\pm0.5}$ | $\mathbf{80.26}_{\pm0.6}$ | $\mathbf{82.64}_{\pm0.4}$ |

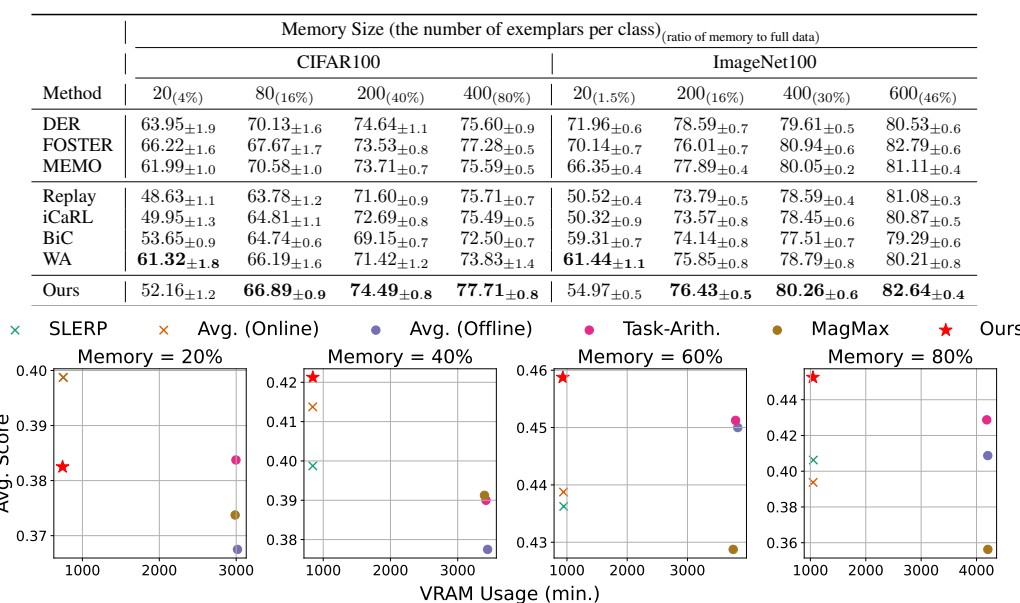

Figure 3: Comparison of average score and relative VRAM usage measured as minutes under different exemplar memory sizes in LLM continual instruction tuning for 8-task using TRACE.

**Continual instruction tuning results.** We evaluate our method in a more practical setting of continual learning for foundation models, where exemplar memory is abundant. To this end, we conduct continual instruction tuning on the TRACE benchmark, following its standard 8-stage setup across diverse domains. As strong baselines, we compare with recent model merging approaches that have been actively explored for continual learning with foundation models (Roth et al., 2024; Dziadzio et al., 2024). Among them, methods such as Task Arithmetic, MagMax, and Model Soup (*i.e.*, Avg. (Offline)) require storing all $T$ task-specific models or representations and merging them post hoc. In contrast, methods like Ours, SLERP, and Avg.(Online) operate in a single model trajectory and do not require saving all intermediate checkpoints. We denote $|W|$ as the standard VRAM usage of a single model. To ensure a fair evaluation, we follow the protocol proposed by (Cha & Cho, 2025): all hyperparameters are tuned only under the 20% memory setting, and the same configuration is reused across other memory sizes without additional tuning. Figure 3 shows that our method consistently outperforms the baselines when the exemplar memory exceeds 20% (Also see Figure 10). In particular, we observe that our method achieves 2–9% higher accuracy across all memory sizes compared to offline merging approaches like Task Arithmetic and MagMax, which require a relative VRAM usage of $|W| * T$ for saving all previous task models in the GPU. Notably, as summarized in Appendix Table 9, these offline merging methods struggle to integrate knowledge from diverse tasks (Dziadzio et al., 2024) and tend to underperform under abundant exemplar memory conditions. Conversely, our method avoids these issues by operating directly in weight space during training and achieves both high performance and GPU efficiency (relative VRAM usage: $|W| * 2$). More experimental results or detailed numerical experimental results are introduced in Appendix A.4.

## 5.3 ABLATION STUDY

We conduct ablation studies to validate the effectiveness of two key components of our method. (1) ranking-based parameter reset for plasticity recovery, and (2) weight averaging for stability.

**Contribution of each component.** Table 3 summarizes the performance of our method when

Table 3: Ablation study on CIFAR-100

| | Memory Size | | | | | | | |
| Method | 20 | 40 | 80 | 100 | 200 | 300 | 400 | 500 |
|---|---|---|---|---|---|---|---|---|
| Replay | 48.92 | 56.59 | 63.30 | 65.12 | 70.84 | 73.46 | 75.38 | 76.90 |
| w/o reset | 50.23 | 58.19 | 65.01 | 66.50 | 72.33 | 75.00 | 76.98 | 77.92 |
| w/o avg. | 48.73 | 56.89 | 63.43 | 65.22 | 70.81 | 73.49 | 74.99 | 76.47 |
| Ours | **52.00** | **59.69** | **66.51** | **67.73** | **73.57** | **76.11** | **77.42** | **78.25** |

either reset or averaging is ablated. The baseline Replay uses neither. We find that weight reset alone (w/o avg.) yields limited gains over Replay, while averaging without reset (w/o reset) improves stability but fails to fully adapt to new tasks. Only when both are combined do we observe substantial gains, highlighting that the two operations are complementary and that weight reset primarily serves to enable effective averaging.

**Effectiveness of the importance metric.** Table 4 compares alternative parameter importance metrics used for reset. Our moment-based metric (Eq. 4) performs on par with expensive Hessian-based scores, while being significantly more efficient. Simple metrics like parameter drift also perform well, echoing prior work in pruning (Zhu & Gupta, 2017; Liu et al., 2018). In contrast, using only the first or second moment leads to poor selection, confirming the necessity of combining both for robustness.

Table 4: Study of Parameter-Importance Metric on Average class-IL accuracy (%) in CIFAR-100

| Metric | Cost | Memory Size | | |
|---|---|---|---|---|
| | | 20 | 200 | 500 |
| Param. Drift | ⇓ | 51.93 | 73.02 | 77.90 |
| Fisher-based | ⇑ | 52.35 | 73.49 | 77.57 |
| Hessian-based | ⇑ | **52.81** | **73.63** | 78.16 |
| First Moment | ⇓ | 50.10 | 71.38 | 75.05 |
| Second Moment | ⇓ | 47.29 | 70.44 | 73.89 |
| Ours (eq. (4)) | ⇓ | 52.00 | 73.57 | **78.25** |

**Reset strategies: how and when to reset.** Finally, we investigate two design choices in the reset operation: the reset rule (how to reset) and the reset frequency (when to reset). As shown in Table 5, our soft reset method (a weighted blend with $\theta_{\mathrm{prev}}$) consistently outperforms random reinitialization and hard reversion. This advantage becomes more pronounced with larger memory. We also compare with existing methods (e.g., Shrink & Perturb (Ash & Adams, 2020) and Continual Backprop. (Dohare et al., 2024)), where our method outperforms competitors. Regarding frequency, performing a reset only once after $n_{\mathrm{warm}}$ works well in most settings. However, in constrained memory regimes, applying multiple resets yields further gains—consistent with findings in sparse reinitialization (Frankle & Carbin, 2019). We provide further

Table 5: Study of reset strategies on Average class-IL accuracy (%) in CIFAR-100

| Strategy | Memory Size | | |
|---|---|---|---|
| | 20 | 200 | 500 |
| *Reset Method* | | | |
| Random | 45.30 | 71.56 | 75.11 |
| Revert | 50.93 | 71.85 | 75.78 |
| Shrink&Perturb 3 | 48.26 | 70.32 | 76.05 |
| Continual Backprop. 18 | 47.15 | 70.49 | 76.52 |
| Ours (w/o Avg.) | 48.73 | 70.81 | 76.47 |
| Ours | **52.00** | **73.57** | **78.25** |
| *Reset Frequency* | | | |
| Every Iter. | 47.38 | 63.11 | 70.32 |
| Every Epoch | **53.36** | 72.73 | 76.57 |
| Once (Ours) | 52.00 | **73.57** | **78.25** |

analysis in Appendix A.4 on how tuning the reset frequency and retain rate $Q$ affects performance.

**Continual Learning vs. Full Retraining.** Prior work (Dohare et al., 2024) reports that with *full* exemplar memory, retraining from scratch (i.e., joint-training) can surpass continual learning (CL). In Section 3.2, we revisit this observation theoretically and argue that, under full memory, *per-task resets* (reinitializing weights before each task) recover plasticity while preventing forgetting, yielding strong performance. However, in the more realistic *sufficient* memory regime (e.g., 20–40%), retraining from scratch degrades markedly as

Table 6: Comparison of Full Retraining from scratch and Continual Learning under a standard CL setting. Average class-IL accuracy (%) on CIFAR-100 is reported.

| Method | Memory Size (# of exemplars per class) (ratio of memory to full data) | | | | |
|---|---|---|---|---|---|
| | 20 (4%) | 100 (20%) | 200 (40%) | 300 (60%) | 400 (80%) |
| Full Retraining | 46.15 | 64.73 | 69.96 | **76.02** | **78.94** |
| Continual (Replay) | 48.63 | 66.11 | 71.60 | 73.29 | 75.71 |
| Ours | **52.16** | **67.25** | **74.49** | 75.97 | 77.71 |

forgetting is insufficiently addressed. In these settings, algorithms that leverage prior models (e.g., ours) offer clear advantages, as seen in Table 6. Finally, while maximal replay might appear ideal, it is often impractical at scale (e.g., LLMs). Our approach provides a cost-effective alternative that balances performance with efficiency

## 6 CONCLUDING REMARKS

This paper challenges the long-standing assumption in continual learning (CL) that exemplar memory is the primary bottleneck, arguing that in modern deployments, GPU cost is the true constraint. We investigate the consequences of having sufficient memory—a practical regime where forgetting is largely solved but full retraining from scratch remains prohibitively expensive—and demonstrate that the central challenge shifts from mitigating forgetting to overcoming a loss of plasticity. To address this, we propose Weight Space Consolidation, a lightweight method combining parameter resets and weight averaging to navigate this new stability-plasticity trade-off. Our method is empirically validated across image classification and LLM instruction tuning, where it outperforms strong baselines at a fraction of the cost, establishing a scalable and cost-efficient alternative to full retraining. Ultimately, our findings call for a crucial shift in focus for future CL research: from optimizing for unrealistic memory constraints toward designing computationally efficient algorithms for the real-world scenarios of today.

ACKNOWLEDGEMENT

We thank NYU HPC for their generous support and help in computational simulations and experiments. This work was supported by the Institute of Information & Communications Technology Planning & Evaluation (IITP) with a grant funded by the Ministry of Science and ICT (MSIT) of the Republic of Korea in connection with the Global AI Frontier Lab International Collaborative Research. It is also supported in part by National Research Foundation of Korea (NRF) grant [No. 2021R1A2C2007884, No. RS-2025-02263628, No. RS-202300265406] and the Institute of Information & communications Technology Planning & Evaluation (IITP) grants [RS-2021-II212068, RS-2022-II220113, RS-2022-II220959, RS-2021-II211343].

ETHICS STATEMENT

The authors acknowledge and concur with the ICLR Code of Ethics, namely in its pursuit of (1) human well-being, (2) high standards of scientific excellence, (3) consideration for the societal impacts (i.e., harms) of AI, (4) honesty & trustworthiness, (5) fairness, (6) mutual respect for other researchers' works, (7) privacy, and (8) confidentiality.

REPRODUCIBILITY STATEMENT

For reproducibility, we provide the source code, experimental guidelines, and the scripts used in our experiments. Please refer to the README.md file in the supplementary materials on how to reproduce our experiments. We also used a fixed seed setting, which is implemented in the source code. We also include notebook (.ipynb) files to reproduce the figures appearing in our paper. Lastly, in Section 5.1 and Appendix A.5, we thoroughly explain how our method and its experiments are implemented.

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

# A APPENDIX

## A.1 ABUNDANT EXEMPLAR MEMORY REGIME

**Summary.** In the abundant exemplar memory regime, the replay buffer becomes an excellent proxy for the union of past tasks, which (1) mitigates forgetting (see Appendix A.1.1) but (2) reduces the effective learning signal for novel information (task), causing high node-reuse and low plasticity (see Appendix A.1.2). Section 5 empirically quantified this trade-off across different replay ratios on CIFAR-100 and ImageNet-100.

### A.1.1 STABILITY: ABUNDANT EXEMPLAR MEMORY MITIGATES CATASTROPHIC FORGETTING

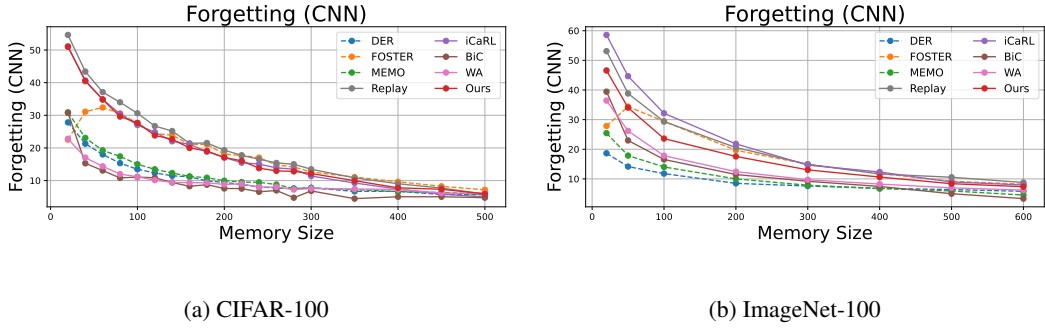

(a) CIFAR-100          (b) ImageNet-100

Figure 4: The impact of exemplar memory size on catastrophic forgetting. Increased memory drastically reduces the forgetting between tasks, while it persists.

Next, we discuss how a sufficiently large exemplar memory buffer $\mathcal{M}$ approximates previous tasks' distributions, thus reducing catastrophic forgetting.

Ideally, if we could train jointly on all tasks $1, \cdots, t$, the obtained model parameter $\theta^*_{1:t}$ would minimize the risk:

$$R_{1:t}(\theta) = \sum_{j=1}^{t} \mathbb{E}_{x \sim P_j}\big[\ell(\theta; x)\big]. \tag{7}$$

which minimizes forgetting by design. By joint-training on all task data up to task $t$, we would find $\theta^*_{1:t} = \arg\min_\theta R_{1:t}(\theta)$, with no conventional notion of forgetting between tasks.

Similarly, with a large enough exemplar memory, the replayed data for previous tasks closely approximates their true distributions ($\tilde{P}_j \approx P_j$ for all $j < t$). Therefore training on $D_t \cup \mathcal{M}_{1:t-1}$ (its distribution eq. (2)) yields a risk

$$\tilde{R}_{1:t}(\theta) \approx \lambda \mathbb{E}_{P_t}\big[\ell(\theta; x)\big] + (1 - \lambda)\sum\nolimits_{j=1}^{t-1} \hat{\pi}_j \mathbb{E}_{\tilde{P}_j}\big[\ell(\theta; x)\big], \tag{8}$$

where $\hat{\pi}_j$ indicates the empirical importance (i.e., size) of the tasks. Here, we may bound $|\tilde{R}_{1:t}(\theta) - R_{1:t}(\theta)| \leq \epsilon$ under standard assumptions (e.g., Lipschitz continuity in $\theta$ (Khromov & Singh, 2024), use of representative samples during empirical risk minimization), with $\epsilon$ shrinking as the replay buffer size $K$ increases, leading to $\tilde{P}_j \approx P_j$. Assuming $R_{1:t}$ is $\mu$-strongly convex in a neighbourhood of $\theta^*_{1:t}$ (i.e., locally strong convex), this risk gap implies $\|\tilde{\theta}^*_{1:t} - \theta^*_{1:t}\| \leq \sqrt{2\epsilon/\mu}$ (Fornasier et al., 2021; Escande, 2024). Intuitively, if the two risk surfaces are proximate, their minimizers are also close in the parameter space (Beer et al., 1992; van de Geer & Wainwright, 2017).

In this sense, forgetting of a previous task $j$ arises when $\tilde{\theta}^*_{1:t}$ drastically changes its predictions on the previous task distribution $P_j$. But if $\tilde{\theta}^*_{1:t}$ remains near $\theta^*_{1:t}$ (which, by definition, does well on previous tasks by training jointly), it must still perform well on task $j$. Hence, if we measure forgetting in the parameter space as

$$\Delta_{j \to t} = \mathbb{E}_{x \sim P_j}\big[\ell(\tilde{\theta}^*_{1:t}; x) - \ell(\theta^*_{1:j}; x)\big], \tag{9}$$

the measure of how replay-based parameters after task $t$ perform on task $j$ compared to the parameters after task $j$. Here, $\Delta_{j \to t}$ remains small if $\tilde{\theta}^*_{1:t} \approx \theta^*_{1:t}$. Naturally, since $\theta^*_{1:t}$ is a reliable minimizer on

all tasks, the small parameter drift ensures that the performance of the model trained under abundant exemplar memory does not degrade on earlier tasks - i.e., forgetting is reduced as exemplar memory becomes larger (Merlin et al., 2022; Brignac et al., 2023).

Experimentally, Figure 4 validates that increasing the exemplar memory size can reduce forgetting (hence improving stability). Here, forgetting is measured as the average over all previously learned tasks of the drop from each task's best-ever accuracy to its accuracy after the final task (Zhou et al., 2023a). Formally:

$$\text{Forgetting} \; = \; \frac{1}{T-1} \sum_{k=1}^{T-1} \Big( \max_{1 \le i \le T} a_{k,i} \; - \; a_{k,T} \Big),$$

where we denote $a_{k,i}$ and $a_{k,T}$ as the accuracy on task $k$ immediately after learning task $i$, and the accuracy on task $k$ after learning the final task $T$, respectively.

### A.1.2 PLASTICITY: ABUNDANT EXEMPLAR MEMORY DETERIORATES MODEL'S CAPACITY TO LEARN NEW TASKS

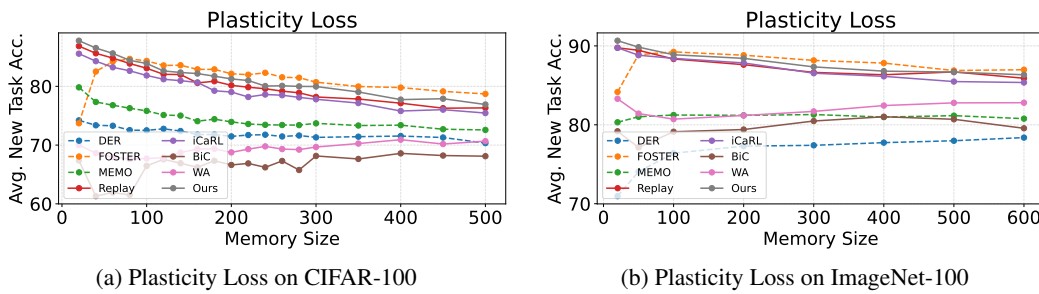

(a) Plasticity Loss on CIFAR-100      (b) Plasticity Loss on ImageNet-100

Figure 5: A comparison of plasticity loss (measured using the average of the new task accuracy) across different exemplar memory sizes in the 10 task scenario of CIFAR-100 and ImageNet-100. As memory size increases, models lose their ability to learn new tasks.

$\lambda$

We claim that while abundant exemplar memory improves stability, it inevitably suppresses plasticity (Dohare et al., 2024). Let $\theta^{(t-1)}$ and $\theta^{(t)}$ denote the network parameters right before and right after training on task $t$. Denote by $g_t(x, y) = \nabla_\theta \ell(\theta^{(t-1)}; x, y)$ the per–example gradient when we begin task $t$. Under the hybrid distribution of equation 2, the expected update direction is

$$\underbrace{\mathbb{E}_{x \sim P_{\text{train}}^{(t)}}[g_t(x)]}_{\triangleq \; \bar{g}_t} \; = \; \lambda \, \bar{g}_t^{(\text{new})} \; + \; (1 - \lambda) \, \bar{g}_t^{(\text{past})}, \tag{10}$$

where $\bar{g}_t^{(\text{new})}, \bar{g}_t^{(\text{past})}$ is the gradient mean over $P_t$ and $P_{\text{past}}$, respectively. Define the gradient alignment

$$\rho_t \; = \; \frac{\langle \bar{g}_t^{(\text{new})}, \bar{g}_t^{(\text{past})} \rangle}{\|\bar{g}_t^{(\text{new})}\| \, \|\bar{g}_t^{(\text{past})}\|} \; \in [-1, 1], \tag{11}$$

using the cosine similarity (Du et al., 2018; Lee et al., 2021). When the buffer is abundant, $P_{\text{past}}^{(t)}$ (i.e., past task data at task step $t$ stored in the replay memory) and $P_{\text{train}}^{(t-1)}$ (i.e., the mixed train data at task step $t - 1$) are close, and by definition the distribution $P_{\text{train}}^{(t)}$ at task $t$ is similar to the distribution $P_{\text{train}}^{(t-1)}$ at task $t - 1$; hence $\rho_t \approx 1$. Consequently, the effective step taken during task $t$

$$\|\theta^{(t)} - \theta^{(t-1)}\| \; = \; \eta \, \|\bar{g}_t\| \; \le \; \eta \big[ \lambda + (1 - \lambda) \big] \|\bar{g}_t^{(\text{new})}\| \; = \; \eta \, \|\bar{g}_t^{(\text{new})}\|, \tag{12}$$

where $\eta$ is the learning rate. Here, Equation (12) shrinks monotonically with $\lambda$ because $\bar{g}_t^{(\text{new})}$ and $\bar{g}_t^{(\text{past})}$ are almost colinear. In the limit $\lambda \to 0$ (i.e. the current data is overwhelmed by replay) the update direction collapses onto the span of previous gradients (Yu et al., 2020; Kendall et al., 2018), yielding

$$\|\theta^{(t)} - \theta^{(t-1)}\| \; \xrightarrow[\lambda \to 0]{} \; 0, \tag{13}$$

which encourages the network to remain dormant (i.e., loss of plasticity (Dohare et al., 2024)) between tasks.

This analysis connects with the *node-reuse* phenomenon studied in (Lee et al., 2022), which claimed that when two sequential tasks are similar–as in the case of $P_{\text{train}}^t$ and $P_{\text{train}}^{(t-1)}$ under the abundant exemplar memory regime–models tend to reuse their nodes (i.e., loss of plasticity), and also aligns with the model's underperformance in training under extreme replay (i.e., full data is provided as exemplar memory) as reported in (Rolnick et al., 2019).

The results of Figure 5 depict how a model's plasticity, more specifically its ability to learn new tasks, diminishes as exemplar memory becomes abundant. Here, the plasticity was measured using the model's average accuracy on new tasks. In Section 3.2, we also provided another analysis focused on the train loss, where we observe that under abundant exemplar memory, convergence becomes difficult. In the sequel, we investigate this phenomenon more deeply.

### A.1.3 DISCUSSION: NEW CHALLENGES UNDER ABUNDANT EXEMPLAR MEMORY

Using exemplar memory results in a hybrid training dataset that combines current task data $D_t \cup \mathcal{M}_{1:t-1}$ with replayed samples from previous tasks.

This combined dataset introduces several side effects. First, as discussed in Section 3.2, stability increases, hence reducing forgetting (Appendix A.1.1). On the other hand, plasticity worsens (Appendix A.1.2). In this section, we investigate whether the abundant exemplar memory regime poses additional issues.

First, gradient interference may occur as the gradients computed on current task samples can conflict with those from past tasks, leading to partial cancellation of updates and impeding effective learning across tasks. We view that this potentially could lead to a new form of forgetting that distorts the learned features and ultimately interfering with the model's learning process. Second, the heterogeneous nature of the hybrid dataset increases the variance of the stochastic gradient estimates, resulting in slower convergence. This heightened variance necessitates either more iterations or more sophisticated optimization techniques to minimize the loss reliably. Furthermore, imbalances in task representation can arise if the replay buffer unevenly captures the diverse distributions of past tasks.

Conventional continual learning methods typically address forgetting by assuming that new data is markedly different from prior data and focusing on preserving performance on previously learned tasks. However, in the abundant exemplar memory regime, where all tasks are presented simultaneously within a combined dataset, these methods fall short. They are not equipped to handle the multi-task learning dynamics and the associated balancing issues that emerge when the model must integrate and harmonize learning signals from a diverse set of tasks. We believe a more thorough investigation is required, and we set this as a key objective of our future work.

### A.2 METHOD (CONTINUED.)

In this section, we elaborate on our method *weight space consolidation*, highlighting the role of each component. The proposed method is a combination of two weight-space operations. (1) ranking-based parameter reset (Yadav et al., 2024) and (2) weight averaging (Wortsman et al., 2022).

**Ranking-based weight reset.** The weight reset step selects and resets redundant parameters in adapting to the new task. The aim of this procedure is to find the minimal set of parameters that are capable of adapting to the new $t^{th}$ task, and reinitializing the redundant parameters to the mixed value of the previous task model $\theta_{prev}$ and the current model $\theta$ value using Equation (5), which helps recover learned features. In this process, we use a simple metric Equation (4) that uses the parameter's first and second moments to measure its importance (Kingma & Ba, 2017; Balles & Hennig, 2020; Molchanov et al., 2019;

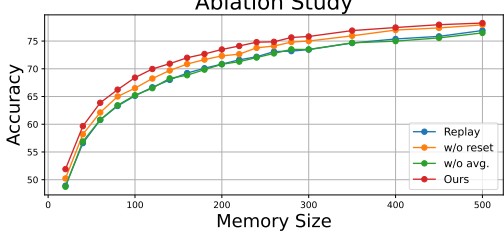

Figure 6: Ablation Study on CIFAR-100. We empirically find that resetting and averaging collaboratively benefit each other.

Hwang, 2024). In Appendix A.3, we also compare alternative measures (e.g., parameter drift) and discuss their limitations.

Using this metric, we retain the top-$Q\%$ parameters that have largely drifted during the current task and reset the dormant parameters using eq. (5). The idea of resetting model weights is not novel in the continual learning literature, but most have focused on improving plasticity (Dohare et al., 2024). Conversely, we improve both plasticity and stability using a mixing technique (eq. (5), integrating previous and present tasks. In appendix A.4, we provide experimental results on the effect of $Q$, the percentage of weight reset on model performance (see fig. 11). Furthermore, we study alternative variations of our reset method, namely (1) different parameter importance metrics (see Table 4), (2) different reset methods and reset frequencies (see Table 5). We find that resetting more frequently (e.g., per epoch) displays significant gains in performance especially under limited memory settings. We discuss this observation on Appendix A.4.

**Weight Average.**    The weight-averaging step aggregates various learned signals into a single model, allowing for faster convergence and improved generalization (Izmailov et al., 2018). The underlying idea is that the model weights can address catastrophic forgetting, functioning as a substitute for replay memory. Using this, we aim to fill the gap between the full exemplar memory setting (i.e., joint-training) and the abundant exemplar memory setting. Another motive behind this approach is the new challenges (see section 3.1) rooted in training with a mixture of datasets $D_t \cup \mathcal{M}_{1:t-1}$, which generally requires more training steps for convergence. In Appendix A.4, we empirically observe that the abundant exemplar memory setting complicates training, requiring more training epochs for convergence (see Figure 2b). The weight averaging technique is also an emerging practice in the continual learning literature (Marouf et al., 2025; Kozal et al., 2024). However, such works merge the model weights after training (i.e., offline merging (Dziadzio et al., 2024)), which requires the storage of multiple model weights (proportional to the number of tasks). On the other hand, our method uses a moving average model that is updated during training (i.e., online merging).

**Why does Weight Space Consolidation work?**    Our starting point is the abundant memory regime, where abundant exemplars reduce forgetting but drive the optimizer toward sharp, over-specialized minima that harm plasticity. Weight Space Consolidation is designed to counteract this effect with two complementary weight-space operations. First, targeted re-initialization of dormant parameters restores unused capacity, enabling the network to escape locally saturated directions and learn new tasks. Second, in-situ weight averaging biases training toward flatter regions of the loss landscape, a mechanism that prior work has linked to improved generalization and reduced forgetting Izmailov et al. (2018); Cha et al. (2021; 2020); Cho et al. (2024). Our experiments further show that this averaging step induces sparsity in the effective parameter usage, and sparsity is known to enhance plasticity by preventing the model from over-relying on a small set of critical parameters Golkar et al. (2019); Dohare et al. (2024). Together, these effects provide an explanation of why our proposed method can simultaneously recover plasticity and maintain stability in the abundant-memory regime.

## A.3    MEASURING PARAMETER IMPORTANCE

In our work, we measure a parameter's contribution to learning a new task by using a moment-based score (eq. (4)). However, there are several alternative approaches we could take. In this section, we investigate the alternative measures that could be used to measure a parameter's importance to learning a new task.

First, we can simply use the parameter drift to measure a model's contribution to the new task, formulated as:

$$\Delta_l = \big|\theta[l] - \theta_{prev}[l]\big|, \tag{14}$$

where $\theta[l]$ and $\theta_{prev}[l]$ indicates the $l^{th}$ parameter of the current model $\theta$ and the previous task model $\theta_{prev}$, respectively. However, a problem with this approach is *when* the previous model $\theta_{prev}$ should be stored. This is a critical issue in cases we wish to reset *multiple* times.

A more principled alternative weights each parameter by the empirical Fisher diagonal:

$$F_l \;=\; \frac{1}{N}\sum_{n=1}^{N}\Big(\partial_{\theta[l]}\log p(y_n \mid x_n;\theta)\Big)^2, \tag{15}$$

which captures how strongly the log-likelihood reacts to perturbations of $\theta[l]$. This idea underpins many existing continual learning methods (e.g., EWC (Kirkpatrick et al., 2017), MAS (Aljundi et al., 2018)). However, Fisher scores are usually difficult to compute real-time and lack scalability.

Another metric we can use is the Hessian estimate (Yu et al., 2021; Chong et al., 2023). Second-order methods replace $F_l$ by a Hessian diagonal estimate, which we can efficiently obtain with Hutchinson's trick (Hutchinson, 1986):

$$\widehat{h}_l = \frac{1}{K} \sum_{k=1}^{K} (v^{(k)} \odot H v^{(k)})_l, \qquad v^{(k)} \sim \{-1, +1\}^{|\theta|}, \tag{16}$$

where each Hutchinson probe costs a single Hessian–vector product. The absolute curvature $s_l = |\widehat{h}_l|$ can then serve as our importance score. However, similar to the Fisher matrix, Hessians are exceptionally costly to compute, especially for large models. In Section 5, we investigate the efficacy of these metrics as a parameter importance measure.

We can also think of differentiating between how a parameter changes (1) between tasks (2) within a task.

**Inter-task Parameter Drift.**    When transitioning from one task to the next, the change in the $l$-th parameter can be measured as

$$\Delta\theta_{\text{inter}}[l] = \big| \theta_t[l] - \theta_{t-1}[l] \big|,$$

where $\theta_t[l]$ denotes the $l^{th}$ parameter after training on the current task $t$, and $\theta_{t-1}[l]$ represents the $l^{th}$ parameter after training on the previous task $t-1$. A large value indicates that the parameter is highly task-specific, while a small value suggests robustness across tasks.

**Intra-task Parameter Drift.**    Alternatively, we can analyze how a parameter evolves during the training process of a single task. Let $\theta^{(i)}[l]$ denote the value of the $l^{th}$ parameter at the $i^{th}$ iteration during training. Then, the intra-task parameter drift is given by

$$\Delta\theta_{\text{intra}}^{(i)}[l] = \big| \theta_t^{(i+1)}[l] - \theta_t^{(i)}[l] \big|.$$

This measure captures the incremental updates of the parameter as the model optimizes its performance on the current task. By comparing both the inter-task and intra-task parameter changes, we gain a more comprehensive understanding of the role each parameter plays in adapting to new tasks as well as the dynamics of learning within a single task. In our future work, we will seek a more reliable metric to express a parameter's behavior in the weight space.

### A.4 EXPERIMENTAL RESULTS AND ANALYSIS

In this section, we provide the full results of our experiments, namely (1) CIFAR-100, (2) ImageNet-100, and (3) TRACE.

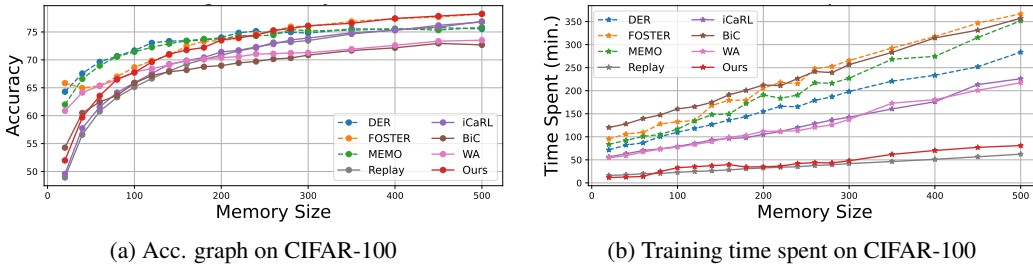

(a) Acc. graph on CIFAR-100                    (b) Training time spent on CIFAR-100

Figure 7: Comparison of (a) average class-incremental accuracy and (b) training time under different exemplar memory sizes in class-incremental learning for 10-task using CIFAR-100. As memory increases, catastrophic forgetting is mitigated, but training time (*i.e.*, computational cost) also grows proportionally. Note that DER, FOSTER, and MEMO are expansion-based methods (shown with dashed lines).

Table 7: Average class-IL accuracy (%) on CIFAR-100 with varying exemplar-memory sizes. We report experimental results with varying memory sizes, ranging from 20 exemplars per class (a common setting in class-IL) to 500 exemplars per class (storing the entire previous dataset in CIFAR-100. Bold highlights the best non-expansion method. We report the standard error across 5 runs.

| Method | Memory Size (the number of exemplars per class)(ratio of memory to full data) | | | | | | | |
|---|---|---|---|---|---|---|---|---|
| | $20_{(4\%)}$ | $40_{(8\%)}$ | $80_{(16\%)}$ | $100_{(20\%)}$ | $200_{(40\%)}$ | $300_{(60\%)}$ | $400_{(80\%)}$ | $500_{(100\%)}$ |
| DER | $63.95_{\pm1.9}$ | $67.27_{\pm1.5}$ | $70.13_{\pm1.6}$ | $70.98_{\pm1.3}$ | $74.64_{\pm1.1}$ | $75.05_{\pm1.3}$ | $75.60_{\pm0.9}$ | $75.92_{\pm1.1}$ |
| FOSTER | $66.22_{\pm1.6}$ | $65.58_{\pm1.5}$ | $67.67_{\pm1.7}$ | $69.01_{\pm1.2}$ | $73.53_{\pm0.8}$ | $76.19_{\pm0.7}$ | $77.28_{\pm0.5}$ | $78.07_{\pm0.7}$ |
| MEMO | $61.99_{\pm1.0}$ | $66.59_{\pm1.1}$ | $70.58_{\pm1.0}$ | $71.44_{\pm0.8}$ | $73.71_{\pm0.7}$ | $75.20_{\pm0.8}$ | $75.59_{\pm0.5}$ | $75.83_{\pm0.5}$ |
| Replay | $48.63_{\pm1.1}$ | $56.80_{\pm1.4}$ | $63.78_{\pm1.2}$ | $66.11_{\pm1.2}$ | $71.60_{\pm0.9}$ | $73.29_{\pm0.5}$ | $75.71_{\pm0.7}$ | $77.02_{\pm0.5}$ |
| iCaRL | $49.95_{\pm1.3}$ | $57.12_{\pm1.3}$ | $64.81_{\pm1.1}$ | $66.23_{\pm1.2}$ | $72.69_{\pm0.8}$ | $73.63_{\pm0.7}$ | $75.49_{\pm0.5}$ | $76.16_{\pm0.7}$ |
| BiC | $53.65_{\pm0.9}$ | $61.13_{\pm0.8}$ | $64.74_{\pm0.6}$ | $65.59_{\pm0.8}$ | $69.15_{\pm0.7}$ | $71.22_{\pm0.5}$ | $72.50_{\pm0.7}$ | $72.83_{\pm0.7}$ |
| WA | $\mathbf{61.32_{\pm1.8}}$ | $\mathbf{63.87_{\pm1.5}}$ | $66.19_{\pm1.6}$ | $66.90_{\pm1.5}$ | $71.42_{\pm1.2}$ | $72.15_{\pm1.5}$ | $73.83_{\pm1.4}$ | $74.09_{\pm1.2}$ |
| Ours | $52.16_{\pm1.2}$ | $60.34_{\pm1.1}$ | $\mathbf{66.89_{\pm0.9}}$ | $\mathbf{67.25_{\pm1.0}}$ | $\mathbf{74.49_{\pm0.8}}$ | $\mathbf{75.97_{\pm0.6}}$ | $\mathbf{77.71_{\pm0.8}}$ | $\mathbf{78.16_{\pm0.6}}$ |

**CIFAR-100.** The results of the CIFAR-100 class-IL experiment are reported in Table 7. Here, we validate that our method is indeed the strongest among non-expansion methods, and even surpasses expansion-based methods under abundant exemplar memory (see Figure 7a). Specifically, we see that in cases where the exemplar memory size is larger than $16\%$ of the full data, our weight space consolidation method outperforms all non-expansion methods, and begins to match the costly expansion-based methods when the exemplar memory ratio exceeds $40\%$. The true strength of our method lies in its training cost (see Figure 7b), where our method's train time is at par with the cheapest baseline (Replay), while taking one-fifth the time of the expansion-based FOSTER (Wang et al., 2022) and non-expansion-based BiC (Wu et al., 2019).

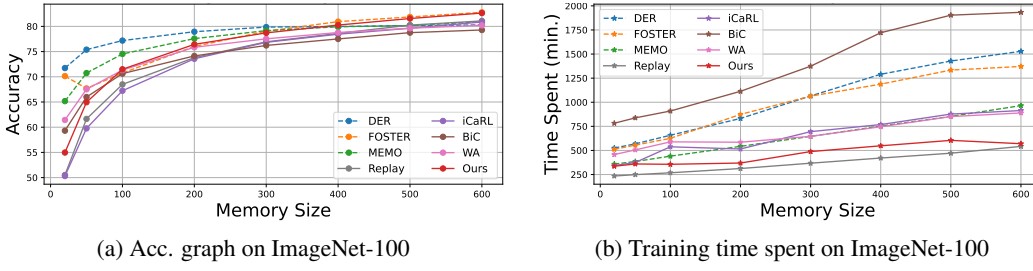

(a) Acc. graph on ImageNet-100          (b) Training time spent on ImageNet-100

Figure 8: Comparison of (a) average class-incremental accuracy and (b) training time under different exemplar memory sizes in class-incremental learning for 10-task using ImageNet-100.

Table 8: Average class-IL accuracy (%) on ImageNet-100 with varying exemplar-memory sizes. Bold highlights the best non-expansion method. We report the standard error across 5 runs.

| Method | Memory Size (the number of exemplars per class)(ratio of memory to full data) | | | | | | | |
|---|---|---|---|---|---|---|---|---|
| | $20_{(1.5\%)}$ | $50_{(4\%)}$ | $100_{(8\%)}$ | $200_{(16\%)}$ | $300_{(23\%)}$ | $400_{(30\%)}$ | $500_{(38\%)}$ | $600_{(46\%)}$ |
| DER | $71.96_{\pm0.6}$ | $75.53_{\pm0.5}$ | $76.80_{\pm0.6}$ | $78.59_{\pm0.7}$ | $79.42_{\pm0.5}$ | $79.61_{\pm0.5}$ | $79.97_{\pm0.6}$ | $80.53_{\pm0.6}$ |
| FOSTER | $70.14_{\pm0.7}$ | $67.69_{\pm0.7}$ | $70.74_{\pm0.5}$ | $76.01_{\pm0.7}$ | $79.03_{\pm0.5}$ | $80.94_{\pm0.6}$ | $81.87_{\pm0.4}$ | $82.79_{\pm0.6}$ |
| MEMO | $66.35_{\pm0.4}$ | $71.12_{\pm0.6}$ | $74.26_{\pm0.4}$ | $77.89_{\pm0.4}$ | $78.74_{\pm0.5}$ | $80.05_{\pm0.2}$ | $80.37_{\pm0.4}$ | $81.11_{\pm0.4}$ |
| Replay | $50.52_{\pm0.4}$ | $61.64_{\pm0.6}$ | $68.49_{\pm0.5}$ | $73.79_{\pm0.5}$ | $76.93_{\pm0.4}$ | $78.59_{\pm0.4}$ | $80.25_{\pm0.5}$ | $81.08_{\pm0.3}$ |
| iCaRL | $50.32_{\pm0.9}$ | $59.76_{\pm0.9}$ | $67.23_{\pm1.0}$ | $73.57_{\pm0.8}$ | $76.84_{\pm0.8}$ | $78.45_{\pm0.6}$ | $79.63_{\pm0.6}$ | $80.87_{\pm0.5}$ |
| BiC | $59.31_{\pm0.7}$ | $65.98_{\pm0.7}$ | $70.63_{\pm0.8}$ | $74.14_{\pm0.8}$ | $76.22_{\pm0.6}$ | $77.51_{\pm0.7}$ | $78.76_{\pm0.6}$ | $79.29_{\pm0.6}$ |
| WA | $\mathbf{61.44_{\pm1.1}}$ | $\mathbf{67.52_{\pm0.9}}$ | $71.33_{\pm1.1}$ | $75.85_{\pm0.8}$ | $77.53_{\pm0.8}$ | $78.79_{\pm0.8}$ | $79.63_{\pm0.9}$ | $80.21_{\pm0.8}$ |
| Ours | $54.97_{\pm0.5}$ | $64.95_{\pm0.6}$ | $\mathbf{71.49_{\pm0.6}}$ | $\mathbf{76.43_{\pm0.5}}$ | $\mathbf{78.71_{\pm0.4}}$ | $\mathbf{80.26_{\pm0.6}}$ | $\mathbf{81.56_{\pm0.4}}$ | $\mathbf{82.64_{\pm0.4}}$ |

**ImageNet-100.** The experimental results in the ImageNet-100 benchmark display a similar pattern. In Figure 8a, we observe a gradual increase in average task accuracy as the exemplar memory size increases, which eventually saturates as it enters the abundant exemplar memory regime. The cost of training increases proportionally to the memory size, where methods like BiC or DER require substantially larger training time. On the other hand, our method displays high accuracy while using roughly one-third    half of the training time (see Figure 8b). For detailed results, please refer to Table 8.

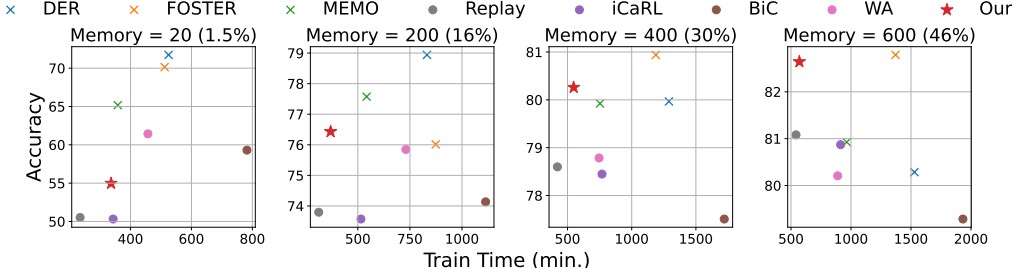

Figure 9: Comparison of (y-axis) average class-incremental accuracy and (x-axis) training time under different exemplar memory sizes in class-incremental learning for 10-task using ImageNet-100. Note that the DER, FOSTER, and MEMO are expansion-based methods (shown with X mark).

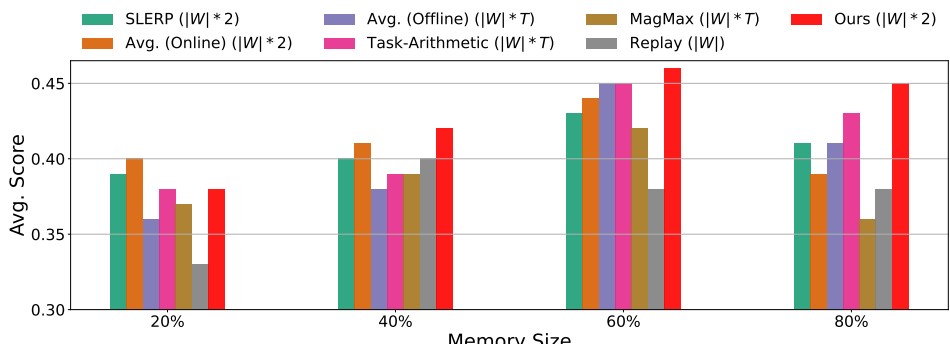

Figure 10: Comparison of average score on different exemplar memory sizes in LLM continual instruction tuning for 8-task using TRACE. The value inside the parentheses indicates the model weight complexity proportional to the number of models loaded in the VRAM.

**TRACE.** TRACE (Wang et al., 2023) is a continual instruction tuning benchmark for the evaluation of LLMs across eight sequential domains, including science (Lu et al., 2022), policy (Shah et al., 2023), meeting summarization (Hu et al., 2023), multilingual classification (Zhao et al., 2023; Gonzales et al., 2021), code generation (Lu et al., 2021), and math reasoning (Mishra et al., 2022).

Table 9 reports the LLM continual instruction tuning benchmark results in TRACE. Similar to the results in the Class-IL benchmarks (e.g., CIFAR-100, ImageNet-100), we observe a similar pattern where the accuracy grows in accordance with the increase in memory size (see Figure 10). An interesting observation is that offline merging (i.e., model merging methods that merge post-training) tends to underperform, displaying lower scores while requiring the VRAM cost of storing multiple models. This aligns with the empirical results of Dziadzio et al. (Dziadzio et al., 2024), which showed that offline merging methods face difficulties in accumulating the multi-task knowledge. Compared to this, our method In addition, note that for TRACE, we use a different method to measure the train cost, which is the relative VRAM usage. The relative VRAM usage is a scaled version of the training time, which measures the computation cost of the training based on the VRAM usage. This scaling is required to distinguish from models that take a similar time to train, but use different numbers of GPUs. For instance, our method and MagMax require similar time to train, but MagMax requires multiple ($t$) task vectors to be loaded in the VRAM, and this is reflected in the relative VRAM usage measure. Considering this, we visualize the results in Figure 3.

**Zero-Memory Setting** To relate our study to exemplar-free continual learning, we also evaluate all methods in a strict zero-memory configuration on CIFAR-100, where no exemplars are stored in the buffer. We report the results in Table 10. This setting corresponds to the idealized regime suggested in some recent CL work, but lies outside our main assumption of exemplar-based CL with sufficient memory. Consistent with our analysis, removing exemplars leads to severe performance degradation across all algorithms: while most methods reach 69–74% average class-IL accuracy in the 40% memory regime (Table 7), their accuracy drops to 25–53% under zero-memory. DER and MEMO achieve the highest performance (53.37% and 43.98%), and our method still improves over Replay (29.83% vs. 26.07%), but the gap to the sufficient-memory regime remains large. These

Table 9: General language understanding and reasoning scores on TRACE with varying exemplar-memory sizes. We report the standard error across 5 runs.

| Method | # Mem. Size | # of $\theta$ stored | Runtime (min.) | C-STANCE Acc. | FOMC Acc. | MeetingBank ROUGE-L | Py150 Similarity | ScienceQA Acc. | NumGLUE$_{cm}$ Acc. | NumGLUE$_{ds}$ Acc. | 20Minuten SAR I | Avg. |
|---|---|---|---|---|---|---|---|---|---|---|---|---|
| Fine-tune | – | – | 137'5 | 0.42 | 0.25 | 0.33 | 0.49 | 0.21 | 0.28 | 0.51 | 0.37 | 0.35 |
| Replay | 0.2 | – | 314'7 | $0.43_{\pm0.08}$ | $0.01_{\pm0.05}$ | $0.35_{\pm0.08}$ | $0.50_{\pm0.07}$ | $0.29_{\pm0.07}$ | $0.32_{\pm0.06}$ | $0.38_{\pm0.05}$ | $0.37_{\pm0.06}$ | 0.33 |
|  | 0.4 | – | 351'2 | $0.49_{\pm0.09}$ | $0.21_{\pm0.07}$ | $0.38_{\pm0.07}$ | $0.50_{\pm0.05}$ | $0.47_{\pm0.04}$ | $0.35_{\pm0.05}$ | $0.41_{\pm0.06}$ | $0.39_{\pm0.05}$ | 0.40 |
|  | 0.6 | – | 394'7 | $0.47_{\pm0.06}$ | $0.13_{\pm0.05}$ | $0.38_{\pm0.05}$ | $0.55_{\pm0.05}$ | $0.18_{\pm0.04}$ | $0.40_{\pm0.03}$ | $0.56_{\pm0.03}$ | $0.37_{\pm0.04}$ | 0.38 |
|  | 0.8 | – | 449'4 | $0.49_{\pm0.03}$ | $0.13_{\pm0.05}$ | $0.37_{\pm0.03}$ | $0.57_{\pm0.04}$ | $0.21_{\pm0.03}$ | $0.38_{\pm0.03}$ | $0.56_{\pm0.03}$ | $0.37_{\pm0.03}$ | 0.38 |
| SLERP | 0.2 | 1 | 375'9 | $0.42_{\pm0.07}$ | $0.45_{\pm0.05}$ | $0.31_{\pm0.06}$ | $0.54_{\pm0.07}$ | $0.36_{\pm0.03}$ | $0.29_{\pm0.05}$ | $0.42_{\pm0.04}$ | $0.40_{\pm0.04}$ | 0.39 |
|  | 0.4 | 1 | 422'4 | $0.45_{\pm0.06}$ | $0.42_{\pm0.06}$ | $0.26_{\pm0.03}$ | $0.53_{\pm0.06}$ | $0.33_{\pm0.05}$ | $0.29_{\pm0.05}$ | $0.48_{\pm0.06}$ | $0.43_{\pm0.06}$ | 0.40 |
|  | 0.6 | 1 | 472'9 | $0.42_{\pm0.06}$ | $0.56_{\pm0.05}$ | $0.38_{\pm0.05}$ | $0.57_{\pm0.03}$ | $0.30_{\pm0.03}$ | $0.34_{\pm0.04}$ | $0.54_{\pm0.05}$ | $0.38_{\pm0.06}$ | 0.43 |
|  | 0.8 | 1 | 526'4 | $0.45_{\pm0.05}$ | $0.18_{\pm0.05}$ | $0.37_{\pm0.06}$ | $0.58_{\pm0.06}$ | $0.39_{\pm0.04}$ | $0.37_{\pm0.03}$ | $0.53_{\pm0.04}$ | $0.38_{\pm0.04}$ | 0.41 |
| Avg. (Online) | 0.2 | 1 | 374'5 | $0.44_{\pm0.05}$ | $0.46_{\pm0.06}$ | $0.33_{\pm0.04}$ | $0.52_{\pm0.05}$ | $0.39_{\pm0.06}$ | $0.31_{\pm0.05}$ | $0.37_{\pm0.05}$ | $0.37_{\pm0.06}$ | 0.40 |
|  | 0.4 | 1 | 421'7 | $0.42_{\pm0.05}$ | $0.41_{\pm0.05}$ | $0.35_{\pm0.05}$ | $0.55_{\pm0.03}$ | $0.43_{\pm0.07}$ | $0.28_{\pm0.03}$ | $0.43_{\pm0.03}$ | $0.44_{\pm0.05}$ | 0.41 |
|  | 0.6 | 1 | 471'4 | $0.45_{\pm0.05}$ | $0.50_{\pm0.05}$ | $0.41_{\pm0.04}$ | $0.55_{\pm0.04}$ | $0.35_{\pm0.05}$ | $0.34_{\pm0.04}$ | $0.52_{\pm0.04}$ | $0.39_{\pm0.05}$ | 0.44 |
|  | 0.8 | 1 | 524'5 | $0.45_{\pm0.03}$ | $0.49_{\pm0.05}$ | $0.35_{\pm0.03}$ | $0.52_{\pm0.03}$ | $0.16_{\pm0.04}$ | $0.35_{\pm0.03}$ | $0.47_{\pm0.05}$ | $0.36_{\pm0.04}$ | 0.39 |
| Avg. (Offline) | 0.2 | $t$ | 377'1 | $0.48_{\pm0.03}$ | $0.02_{\pm0.01}$ | $0.37_{\pm0.03}$ | $0.58_{\pm0.02}$ | $0.29_{\pm0.03}$ | $0.34_{\pm0.05}$ | $0.51_{\pm0.05}$ | $0.37_{\pm0.03}$ | 0.36 |
|  | 0.4 | $t$ | 430'6 | $0.45_{\pm0.04}$ | $0.03_{\pm0.01}$ | $0.38_{\pm0.02}$ | $0.58_{\pm0.05}$ | $0.33_{\pm0.03}$ | $0.40_{\pm0.03}$ | $0.48_{\pm0.03}$ | $0.37_{\pm0.04}$ | 0.38 |
|  | 0.6 | $t$ | 479'3 | $0.44_{\pm0.04}$ | $0.56_{\pm0.03}$ | $0.38_{\pm0.04}$ | $0.59_{\pm0.02}$ | $0.36_{\pm0.03}$ | $0.38_{\pm0.04}$ | $0.53_{\pm0.03}$ | $0.36_{\pm0.01}$ | 0.45 |
|  | 0.8 | $t$ | 525'4 | $0.50_{\pm0.04}$ | $0.21_{\pm0.03}$ | $0.38_{\pm0.03}$ | $0.59_{\pm0.03}$ | $0.35_{\pm0.03}$ | $0.38_{\pm0.02}$ | $0.49_{\pm0.04}$ | $0.37_{\pm0.02}$ | 0.41 |
| Task-Arith. | 0.2 | $t$ | 374'6 | $0.51_{\pm0.03}$ | $0.02_{\pm0.01}$ | $0.39_{\pm0.05}$ | $0.56_{\pm0.05}$ | $0.29_{\pm0.04}$ | $0.40_{\pm0.03}$ | $0.52_{\pm0.04}$ | $0.38_{\pm0.05}$ | 0.38 |
|  | 0.4 | $t$ | 427'6 | $0.49_{\pm0.03}$ | $0.01_{\pm0.01}$ | $0.39_{\pm0.05}$ | $0.59_{\pm0.04}$ | $0.32_{\pm0.05}$ | $0.40_{\pm0.02}$ | $0.53_{\pm0.04}$ | $0.39_{\pm0.03}$ | 0.39 |
|  | 0.6 | $t$ | 474'9 | $0.50_{\pm0.03}$ | $0.47_{\pm0.06}$ | $0.38_{\pm0.05}$ | $0.60_{\pm0.03}$ | $0.32_{\pm0.03}$ | $0.42_{\pm0.03}$ | $0.52_{\pm0.03}$ | $0.40_{\pm0.03}$ | 0.45 |
|  | 0.8 | $t$ | 523'1 | $0.49_{\pm0.03}$ | $0.28_{\pm0.03}$ | $0.39_{\pm0.04}$ | $0.57_{\pm0.04}$ | $0.40_{\pm0.03}$ | $0.44_{\pm0.05}$ | $0.48_{\pm0.02}$ | $0.38_{\pm0.03}$ | 0.43 |
| MagMax | 0.2 | $t$ | 373'3 | $0.49_{\pm0.04}$ | $0.44_{\pm0.03}$ | $0.25_{\pm0.06}$ | $0.26_{\pm0.06}$ | $0.37_{\pm0.04}$ | $0.31_{\pm0.03}$ | $0.48_{\pm0.05}$ | $0.39_{\pm0.04}$ | 0.37 |
|  | 0.4 | $t$ | 424'8 | $0.46_{\pm0.04}$ | $0.41_{\pm0.03}$ | $0.39_{\pm0.05}$ | $0.26_{\pm0.06}$ | $0.35_{\pm0.06}$ | $0.43_{\pm0.04}$ | $0.43_{\pm0.05}$ | $0.40_{\pm0.05}$ | 0.39 |
|  | 0.6 | $t$ | 469'9 | $0.59_{\pm0.05}$ | $0.45_{\pm0.04}$ | $0.37_{\pm0.04}$ | $0.32_{\pm0.06}$ | $0.43_{\pm0.05}$ | $0.42_{\pm0.04}$ | $0.45_{\pm0.03}$ | $0.40_{\pm0.03}$ | 0.42 |
|  | 0.8 | $t$ | 525'6 | $0.32_{\pm0.03}$ | $0.54_{\pm0.05}$ | $0.23_{\pm0.04}$ | $0.25_{\pm0.06}$ | $0.32_{\pm0.03}$ | $0.29_{\pm0.04}$ | $0.51_{\pm0.03}$ | $0.39_{\pm0.04}$ | 0.36 |
| Ours | 0.2 | 1 | 370'9 | $0.39_{\pm0.05}$ | $0.13_{\pm0.03}$ | $0.36_{\pm0.07}$ | $0.51_{\pm0.07}$ | $0.32_{\pm0.04}$ | $0.40_{\pm0.02}$ | $0.56_{\pm0.03}$ | $0.39_{\pm0.01}$ | 0.38 |
|  | 0.4 | 1 | 422'6 | $0.41_{\pm0.03}$ | $0.48_{\pm0.02}$ | $0.38_{\pm0.05}$ | $0.55_{\pm0.05}$ | $0.45_{\pm0.02}$ | $0.41_{\pm0.03}$ | $0.30_{\pm0.03}$ | $0.39_{\pm0.02}$ | 0.42 |
|  | 0.6 | 1 | 465'8 | $0.44_{\pm0.02}$ | $0.57_{\pm0.02}$ | $0.35_{\pm0.04}$ | $0.52_{\pm0.05}$ | $0.47_{\pm0.02}$ | $0.38_{\pm0.02}$ | $0.55_{\pm0.03}$ | $0.39_{\pm0.01}$ | 0.46 |
|  | 0.8 | 1 | 522'9 | $0.46_{\pm0.02}$ | $0.47_{\pm0.02}$ | $0.37_{\pm0.05}$ | $0.55_{\pm0.03}$ | $0.46_{\pm0.02}$ | $0.36_{\pm0.02}$ | $0.53_{\pm0.01}$ | $0.40_{\pm0.01}$ | 0.45 |

results support our claim that exemplar-free CL is substantially more challenging and currently yields accuracy that is insufficient for many real-world deployments, whereas allocating a moderate exemplar buffer enables practical performance gains that our cost-efficient method is designed to exploit.

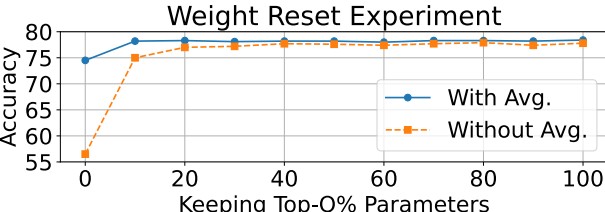

Figure 11: Weight reset experiment on CIFAR-100. Most parameters ($80\%$) are redundant in learning new tasks. Weight Avg. boosts sparsity, retaining performance with fewer ($10\%$) parameters.

**Sparsity analysis after reset.** Next, we examine how much of the model contributes to learning new tasks. In Figure 11, we vary the retain rate $Q$ in the reset step. We find that resetting up to 80% of parameters yields minimal degradation in accuracy, suggesting that only a small subset of weights are actively involved in continual learning. This aligns with findings in sparse training and pruning (Frankle & Carbin, 2019; Chen et al., 2020). We observe that weight averaging helps retain performance under extreme resets (e.g., $Q$=10%), suggesting that it encourages robust, sparse models that do not rely on a small set of critical parameters.

**Effect of Sampling techniques** In this section, we study the effect of batch sampling in exemplar-based continual learning. Prior work has examined the role of exemplar selection and sampling, with early results suggesting that sophisticated schemes such as reservoir sampling may improve performance under severely constrained memory Kim et al. (2020), but more recent large-scale studies report that random

Table 10: Average class-IL accuracy (%) on CIFAR-100 under zero memory.

| Method | Memory Size 0 |
|---|---|
| DER | 53.37 |
| FOSTER | 26.43 |
| MEMO | 43.98 |
| Replay | 26.07 |
| iCaRL | 29.03 |
| BiC | 28.29 |
| WA | 42.10 |
| Ours | 29.83 |

Table 11: Average class-IL accuracy (%) on CIFAR-100 with varying exemplar-memory sizes using reservoir sampling. Memory Size indicates the number of exemplars per class. The value inside the parentheses indicates the gap between the default random sampling and the reservoir sampling results. Bold highlights the best non-expansion method.

| Method | Memory Size (ratio of memory to full data) | |
| --- | --- | --- |
| | $20_{(4\%)}$ | $200_{(40\%)}$ |
| DER | $64.21_{(+0.26)}$ | $74.83_{(+0.19)}$ |
| FOSTER | $65.97_{(-0.25)}$ | $73.71_{(+0.18)}$ |
| MEMO | $62.36_{(+0.37)}$ | $73.40_{(-0.31)}$ |
| Replay | $49.32_{(+0.69)}$ | $71.66_{(+0.06)}$ |
| iCaRL | $50.34_{(+0.39)}$ | $72.89_{(+0.20)}$ |
| BiC | $53.86_{(+0.21)}$ | $69.27_{(+0.12)}$ |
| WA | $\mathbf{61.68}_{(+0.36)}$ | $71.65_{(+0.23)}$ |
| Ours | $52.49_{(+0.33)}$ | $\mathbf{74.63}_{(+0.14)}$ |

Table 12: Average class-IL accuracy (%) on CIFAR-100 with varying exemplar-memory sizes on longer task sequences (T=20). Memory Size indicates the number of exemplars per class. The value inside the parentheses indicates the gap between the default task sequence (T=10; 10 tasks) and the longer task sequence setting (T=20; 20 tasks).

| Method | Memory Size (ratio of memory to full data) | |
| --- | --- | --- |
| | $20_{(4\%)}$ | $200_{(40\%)}$ |
| DER | $55.42_{(-8.53)}$ | $70.91_{(-3.73)}$ |
| FOSTER | $55.94_{(-10.28)}$ | $71.80_{(-1.73)}$ |
| MEMO | $54.61_{(-7.38)}$ | $70.14_{(-3.57)}$ |
| Replay | $44.73_{(-3.90)}$ | $70.88_{(-0.72)}$ |
| iCaRL | $45.28_{(-4.67)}$ | $69.24_{(-3.45)}$ |
| BiC | $45.15_{(-8.50)}$ | $68.97_{(-0.18)}$ |
| WA | $\mathbf{52.36}_{(-8.96)}$ | $70.35_{(-1.07)}$ |
| Ours | $49.77_{(\mathbf{-2.39})}$ | $\mathbf{76.65}_{(\mathbf{+0.22})}$ |

and advanced sampling strategies behave similarly in practice Masana et al. (2022). We expect this trend to persist in the abundant-memory regime considered in our work. To verify this, we re-run all methods on CIFAR-100 using reservoir sampling for exemplar selection, ensuring that the memory buffer fairly represents past tasks. As shown in Table 11, the differences with respect to our default random sampling (values in parentheses) are consistently small, and the gap shrinks further as memory size (i.e., number of exemplars per class in slass-IL) increases from $4\%$ to $40\%$. This indicates that, when sufficient memory is available, the choice of sampling strategy has only a marginal impact on overall performance.

An alternative way to improve plasticity is to enforce a fixed ratio of current-task data within each mini-batch by concatenating current and replay examples at the batch level. While such a design can indeed enhance plasticity, it directly reduces stability: aggressively prioritizing current data accelerates catastrophic forgetting of previous tasks and increases GPU cost due to more constrained batch construction. Thus, batch-level concatenation is complementary but cannot replace our approach, which targets the plasticity–stability trade-off through weight-space operations rather than through carefully engineered batches. Moreover, as highlighted by Dohare et al. (2024), loss of plasticity is a general phenomenon in deep continual learning; our Weight Space Consolidation provides a robust way to restore plasticity that is largely independent of specific batch-design or sampling heuristics.

**Effect of Longer Task Sequences** To assess how our findings scale with the length of the continual learning stream, we further evaluate all methods on a longer task sequence with $T = 20$ tasks on

CIFAR-100, while keeping the total data and exemplar-memory budget fixed. As shown in Table 12, average class-IL accuracy systematically degrades when moving from the default $T = 10$ setting to $T = 20$, and this degradation is particularly pronounced in the low-memory regime (20 exemplars per class), indicating that longer task sequences amplify both forgetting and the loss of plasticity. Nonetheless, in the sufficient-memory setting (200 exemplars per class, i.e., $40\%$ memory), our method maintains strong performance and exhibits a comparatively smaller drop than many baselines, demonstrating that our Weight Space Consolidation remains effective even as the task sequence becomes longer.

**Additional Experiments.** Lastly, we show that under abundant exemplar memory, convergence becomes difficult, similar to a multi-task learning setting. The results are illustrated in Figure 2b. Specifically, we analyze the training loss of the model under two cases. (1) Continual Learning: train the model across 20 tasks sequentially, with abundant exemplar memory. (2) Full retraining (Joint Training): train individual models for each task using all known task data. For this experiment, we used the CIFAR-100 dataset divided into 20 subtasks. In Figure 2b, we can see that under abundant exemplar memory, converging to the training data becomes difficult, especially for the continual learning model. While a simple solution would be to extend the training epochs for convergence, it would collide with our aim for cost-efficiency. On the other hand, joint-trained models relatively converge better. This result aligns with our conjecture on plasticity (see Section 3.2), the results of (Dohare et al., 2024) that a model sequentially trained on different tasks (i.e., continual learning) suffers a loss of plasticity (i.e., a model's ability to learn new tasks), as well as the results of (Rolnick et al., 2019), which demonstrated that learning a new task becomes difficult under extreme (i.e., full ratio) replay memory settings. We believe a thorough investigation of this new phenomenon is required.

## A.5 EXPERIMENTAL DETAILS

In this section, we report the experimental details of our experiments. In all our class-IL experiments, we have used the PyCIL (Zhou et al., 2023a) library, which allows easy replication. We followed the standard training hyperparameters of the PyCIL library, which are fixed across experiments. For the LLM continual instruction tuning experiments, we have used the TRACE (Wang et al., 2023) library. We followed the default training hyperparameters of the TRACE library. Regarding unique hyperparameters, the average interval $j$ was set as 5, and the warming epoch $n_{warm}$ was set as $25\%$ of the total training epochs (default set as 70 under the PyCIL setting). $j$ and $n_{warm}$ were selected using a grid search. The hyperparameters searched in the CIFAR-100 benchmark were applied without modification to the ImageNet-100 experiments. In the LLM experiments, the hyperparameters were selected in the 0.2 ($20\%$) memory setting and applied to the other settings. Note that in the TRACE setting, only one epoch is provided in the replay stage, hence, the average interval was set as 20 iterations, not epochs. Please refer to Section 4 for a better understanding of each hyperparameter. Regarding the model architectures, we used a ResNet-32 for the CIFAR-100 and a ResNet-18 for the ImageNet-100 experiments, following standard settings in PyCIL. For the TRACE experiments, we used a Llama-3.2-1B model. Lastly, regarding the computing resources, we used a single NVIDIA RTX 6000 GPU for all class-IL experiments. For the LLM experiments, we used three V100 GPUs. For our experiments, we used the 2.2.1 version of Pytorch (Paszke et al., 2019).

## A.6 RELATED WORKS (CONTINUED.)

**Weight Space Operations.** Recent works have shown that manipulating model parameters directly with weight-space operations (e.g., model merging (Wortsman et al., 2022)) can handle multi-task learning (Yu et al., 2024) and continual learning (Marouf et al., 2025; Marczak et al., 2025) in a more principled way. These techniques usually intervene post-training by merging the weights of different models e.g., (Yadav et al., 2024) suggested a selective merging algorithm that mitigates the interference between different models, while (Ilharco et al., 2022) showed that arithmetic operations on the weight space can edit the model without training. Unlike these post-training interventions Cho et al. (2025b), our approach manipulates the model's weight space amidst training (Izmailov et al., 2018; Jang et al., 2025) without storing multiple model parameters, aiming for cost-effective editing of the continual learner. Another relevant yet overlooked topic is the effect of weight-space operations on model attributes e.g., loss landscape (Li et al., 2018; Kaur et al., 2023) and plasticity (Dohare et al.,

2024), that contribute to continual learning and generalization. This work empirically investigates various aspects of the model to study their effect on the model's ability to handle distribution shifts. In the continual learning literature, several works have adopted weight-space operations to obtain multi-task solutions without full retraining. For instance, (Kozal et al., 2024) has suggested the use of weight averaging techniques for continual learning, and (Marczak et al., 2025) has extended the idea using task arithmetic. However, these approaches merge models post-training and require the storage of multiple model weights during training. On the other hand, our approach utilizes weight-space operations amidst training, without the redundant storage of multiple model weights. We view this as an important difference in modern settings where the model size is exponentially growing.

## A.7 LIMITATIONS

While our method demonstrates strong performance and cost-efficiency under abundant exemplar memory, it assumes access to a representative subset of past data, which may not always be feasible in privacy-sensitive or streaming-only environments. Additionally, our analysis primarily focuses on class-incremental learning and continual instruction tuning with relatively clean task boundaries. Future work may explore how the proposed weight-space strategies generalize to more complex settings such as task-agnostic CL, online CL, or continual reinforcement learning.

## A.8 SOCIETAL IMPACTS

This work focuses on improving a general capability (e.g., continual learning) of machine learning models, and thus does not directly relate to or cause negative societal impacts. However, we do mention and consider the computational cost of deploying models. We believe that the energy consumption issue of modern machine learning models is an important topic, and in that sense, our work on cost-efficient learning algorithms can indirectly contribute to building a sustainable practice for the training and deployment of artificial intelligence.

## A.9 ASSETS

In our work, many existing assets were used. For the implementation of the models and the learning algorithms, we have used the Pytorch (Paszke et al., 2019) (BSD-3 license) and Huggingface (Wolf et al., 2020) libraries (varies on each library. For instance, the core Transformers library uses an Apache 2.0 license). All of the datasets we have used are public datasets. For instance: CIFAR-100 (MIT License), ImageNet-100 (Custom, Non-commercial Research Only, has a separate terms of use). Regarding the language datasets in TRACE: C-STANCE (CC BY-NC 4.0) ), FOMC (CC BY-NC-SA 4.0), MeetingBank (CC BY-NC-SA 4.0), Py150 (MIT License), ScienceQA (MIT License), NumGLUE-cm (MIT License), NumGLUE-ds (MIT License), and 20Minuten (CC BY-NC-SA 4.0).

