# OpenReview forum: "Forget Forgetting: Continual Learning in a World of Abundant Memory"
_ICLR.cc/2026/Conference — ICLR 2026 Poster_

### Official Review · Reviewer_nQUU · 2025-10-21

**Soundness:** 3
**Presentation:** 4
**Contribution:** 2
**Rating:** 2
**Confidence:** 3

**Summary:**

Traditionally, continual learning has been studied under the memory-restricted setting (i.e., completely exemplar-free or only limited buffer sizes allowed). Authors argue that this is unrealistic under modern standards: compute, and not memory-costs, are the main drivers for the high costs related to model training. Thus, authors study the setting where larger-than-common buffer sizes are used, and where instead the compute availability is the driving factor for comparison. They find that, as the memory buffer grows and thus allows storing more samples of prior tasks, the challenge in CL shift from forgetting to plasticity. Models are stable wrt old tasks, but not plastic enough to new tasks. To alleviate, they propose using stochastic weight averaging and resetting weights with relatively little gradient activity. They study their method, termed Weight Space Consolidation, under the class-incremental setting on CIFAR100, ImageNet-100, and text-based TRACE benchmark, where competitors are outperformed.

**Strengths:**

Strengths:
+ novel take on memory buffers in CL
+ simple and effective method to improve plasticity under performance-stable regimes
+ image and text modalities tested
+ easy-to-follow description of algorithm

**Weaknesses:**

- l 311 to 319: this is a central part of the proposed method, but no experimental results support this thesis
- Paper claims less compute overhead from their method, but no tabular results indicating the overhead versus naive replay are given
- if we can store samples of old tasks, then we could, in turn, learn longer task sequences. Paper should integrate longer task sequences (e.g., TinyImageNet split into 20 or 40 tasks)
- Full re-training is often used as comparison, but missing from the tables
- Its unclear how "from scratch" is interpreted. Is that an entirely fresh model that's trained on increasing unions of memory buffer? Then, no knowledge would be transferred by design
- Hyperparameter tuning on ResNet32 and then using tuned params on ResNet18 seems strange (l 364). Please elaborate on this.
- I do not agree with the definition of post-hoc merging versus in-training merging (l292ff). Post-hoc merging would be, in my understanding, merging after Training on a task, which is thus before a next task -- which is still "during training".

**Questions:**

I have several questions/ideas for improvement
- You use a rank-based approach --> where are the ranks (i.e., matrix ranks) used here? Do you mean "ranking based approach"?
- Equation 2 and 5 both use $\alpha$ for different jobs
- Equation 2: how is alpha annealed (l 232)
- Equation 5: describe that the [l] are the weights to be reset
- Figure 1: are the memory sizes per class or total? Additionally give percentage
- L 460: Q versus q (in the top-q%)
- Figure 3 and 405ff: how is VRAM usage measured? Do not modern frameworks pre-allocate as much GPU memory as possible (i.e., aiming at 100% GPU memory utilization)

---

> ### Author Response · Authors · 2025-11-19
>
> We thank you for your time and effort in reviewing our paper. We appreciate the constructive feedbacks and for acknowledging the simplicity of our method and its effectiveness in multiple modalities. Below, we responded to your concerns one by one.

---

> ### Author Response · Authors · 2025-11-19
> **[for Reviewer nQUU] Response to Weaknesses 1-4**
>
> ## Weaknesses
>
> ***
> ### W1. No experiments to support 'weight averaging improves CL'
> > "l 311 to 319: this is a central part of the proposed method, but no experimental results support this thesis"
>
> We thank the reviewer for pointing this out. In lines 311-319, we do not introduce a new empirical claim. The cited lines state that diverse, heterogenous data can increase the variance of gradient estimates and worsening convergence as observed in prior works [4]. We reconfirmed this in Sec 3.2 (see Fig 2(b)) by showing that under abundant exemplar memory (=hence with heterogeneous data), convergence becomes difficult. In the revision, we will rephrase this passage to make the message clear.
>
>
> Furthermore, the cited papers [1,2] alongside many others [3] have already shown theoretically and empirically that weight averaging improves stability and generalization. This is also supported throughout our experiments (see Tab. 3 -- w/o reset. We showed that averaging alone helps improve CL).
>
> ***
> ### W2. No tabular results indicate less compute overhead of our method (vs. naive replay)
>
> We would like to suggest the figures in our manuscript, such as Fig 1, 3 and 9. **The x-axis of the figures (i.e., the train time ) indicate the GPU cost under a fixed same memory setting**. Here, we confirm that our method requires very GPU low cost showing highest cost-efficiency while achieving superior performance, compared to other SOTA baselines.
>
> ***
> ### W3. Paper should integrate longer task sequences (e.g., 20 or 40 tasks)
>
> We agree that reporting the model's CL behavior on longer task sequences can be insightful. Reflecting the reviewer's suggestion, we have added additional experimental results below.
>
> Method | Memory 20 (4%) | Memory 200 (40%)
> ---|---|---
> Replay | 44.73 (-3.90) | 70.88 (-0.72)
> Ours | 49.77 **(-2.39)** | 74.63 **(+0.14)**
> WA | 52.36 (-8.96) | 70.35 (-1.07)
> iCaRL | 45.28 (-4.67) | 69.24 (-3.45)
> DER | 55.42 (-8.53) | 70.91 (-3.73)
> MEMO | 54.61 (-7.38) | 70.14 (-3.57)
> FOSTER | 55.94 (-10.28) | 71.80 (-1.73)
> BiC | 45.15 (-8.50) | 68.97 (-0.18)
>
>
>
> *Average class-IL accuracy (%) on CIFAR-100 for T=20 tasks. Values inside parentheses denote the absolute change in accuracy (%) relative to the corresponding T=10 setting. Negative values indicate performance degradation when increasing the number of tasks*
>
>
> Our results show that under longer task sequences, both stability and plasticity become worse than the T=10 case. For instance, in the CIFAR-100 setting with 20 tasks (T=20), average class-IL accuracy consistently drops compared to the default setting (T=10), especially in the low-memory regime, indicating that longer task sequences amplifies both forgetting and plasticity loss. In spite of this, **our method with sufficient exemplar memory (i.e., Memory 200 (40%)) shows strength in retaining performance even under longer task sequences**. We will include these extended results in the appendix and reference them in the main text.
>
>
> ***
> ### W4. Full re-training == From Scratch
> > "Full re-training is often used as comparison, but missing from the tables"
>
> We confirm that the results for this baseline, referred to as '**From Scratch', are already provided in Table 6**. We clarify that **'Full Retraining' is typically utilized not as a standard CL baseline**, but rather to empirically demonstrate the effect of plasticity loss, particularly in very large to full memory regimes. Conversely, **Replay method (resuming training from the prior model checkpoint using current task data plus exemplars) serves as the widely adopted naive baseline in both PyCIL and TRACE benchmarks**, which we followed. To enhance clarity, we will update the paper and show this result in both Tab.2 and Tab. 6 of the manuscript.
>
> Method | 20 (4%) | 100 (20%) | 200 (40%) | 300 (60%) | 400 (80%)
> ---|---|---|---|---|---
> From Scratch / Full re-training | 46.15 | 64.73 | 69.96 | 76.02 | 78.94
> Continual (Replay) | 48.63 | 66.11 | 71.60 | 73.29 | 75.71
> Ours | 52.16 | 67.25 | 74.49 | 75.97 | 77.71

---

> ### Author Response · Authors · 2025-11-19
> **[for Reviewer nQUU] Response to Weaknesses 5-7**
>
> ### W5. 'From Scratch'
>
> > "Its unclear how "from scratch" is interpreted", "Is that an entirely fresh model that's trained on increasing unions of memory buffer? "
>
> 'From Scratch' in Tab. 6 is identical to the 'full re-training' baseline that uses an entirely fresh model (the initialization is same across tasks) for each task.
>
> > "Then, no knowledge would be transferred by design"
>
> Exactly, *from scratch* or *full re-training* transfers no knowledge because the model is re-initialized per task. Ironically, this simple re-initialization technique was reported to address the *loss of plasticity* under *very large exemplar memory* [1] (i.e., its effect decreases under more realistic, small~abundant memory scenarios). Again, this method is not generally considered as a standard baseline for CL, as discussed above.
>
> ***
> ### W6. Hyperparameter-tuning on one benchmark and transfering to another
>
>
> Thank you for the question. We followed the evaluation protocol from [6], which is designed to assess CL algorithm's generalizability from a seen dataset (used for tuning) to an unseen dataset (used for evaluation).
>
> First, we performed hyperparameter optimization for all methods on CIFAR-100, which served as our "seen" scenario. Then, we applied these optimal hyperparameters directly to ImageNet-100 (the "unseen" scenario) for the final evaluation. This two-stage process allows for a more rigorous assessment, as it tests how well an CL algorithm's capacity generalizes beyond the dataset it was tuned on, mimicking a real-world application of CL algorithms.
>
> ***
> ### W7. Definition of post-hoc merging versus in-training merging
>
>
> Thank you for your comment. We use the term "in-situ" as it is commonly differentiated from "post-hoc" in merging-based CL works [7]. "Post-hoc" typically refers to methods that merge separate, fully-trained models (e.g., soup[8]-like merging of already trained models). In contrast, our method is in-situ: we perform a merge during the training process (e.g., merging of models during a task training). This merged model is then used as the initialization for the subsequent task, directly integrating the merging process into the learning pipeline. Whereas other methods *post-merge* different task models after training.

---

> ### Author Response · Authors · 2025-11-19
> **[for Reviewer nQUU] Response to Questions and References**
>
> ## Questions
>
> ***
> ### Q1. rank-based approach vs. ranking-based approach
>
> Thank you for pointing this out, you are right. The correct term should be 'ranking-based' (as mentioned in l274). We will correct it in the revised paper.
>
> ***
> ### Q2. The two alphas $\alpha$ and $\alpha_{t}$
> ### Q3. Eq.2.: How is $\alpha$ annealed
>
> Thank you for **both** of your comments. To clarify, $\alpha$ in Eq. 5 and $\alpha_{t}$ in Eq. 2 were notated differently.
>
> $\alpha$ : the weight factor for soft re-initialization ; our algorithm's hyperparameter. **It is fixed as $0.5$ and not annealed.**
>
> $\alpha_{t}$: the exemplar memory size. This parameter defines the exemplar memory setting and **not our algorithm's hyperparameter**. e.g., if there is no exemplar memory, then 1.
>
> We will update the notations in our revised version. Thank you again for pointing this out.
>
> ***
> ### Q4. describe that the [l] in Eq.5 are the weights to be reset
>
> Thank you, $l$ was described in line 275 (in the same page as Eq. 5). $l$ indicates the $l$-th parameter element. We will also remind this near Eq. 5 to make it more visible.
>
> ***
> ### Q5. Clarification regarding Fig. 1
> > "Figure 1: are the memory sizes per class or total? Additionally give percentage"
>
> The 'memory sizes' indicate the number of samples per class. The term follows the term commonly used in class-incremental PyCIL setting. We will clarify this in the caption of Fig. 1. Furthermore, we thank the reviewer for their suggestion on percentages. Since Tab 2 contains the percentages, we will update Fig. 1 accordingly.
>
> ***
> ### Q6. L 460: Q versus q (in the top-q% *in line 286*)
>
> Thank you for finding this typo. We used $Q$ throughout our paper. We will make sure the notation is updated.
>
> ***
> ### Q7. Figure 3 and 405ff
>
>
> We appreciate the question and agree that the measurement protocol should be clearer. The VRAM usage indicates the amount of resource required in storing each task model for post-hoc merging techniques. Our method is computationally efficient as it is not required to load $t$ individual task models. Empirically, we observed that our method runs on smaller GPU resources while other baselines cannot.
>
>
> ***
> ### [References]
>
> [1] Izmailov et al., Averaging weights leads to wider optima and better generalization, UAI (2018).
>
> [2] Cha et al.,  Cpr: classifierprojection regularization for continual learning, ICLR (2021).
>
> [3] Cha et al.,  SWAD: Domain Generalization by Seeking Flat Minima, NeurIPS (2021).
>
> [4] Balles et al., Coupling Adaptive Batch Sizes with Learning Rates, NeurIPS (2021).
>
> [5] Dohare et al., Loss of Plasticity in Deep Continual Learning, Nature (2024).
>
> [6] Cha et al., Hyperparameters in Continual Learning: A Reality Check, TMLR (2025).
>
> [7] Dziadzio et al., How to Merge Your Multimodal Models Over Time?, CVPR (2025).
>
> [8] Wortsman et al., Model soups: averaging weights of multiple fine-tuned models improves accuracy without increasing inference time, ICML (2022).

---

> > ### Comment · Reviewer_nQUU · 2025-11-21
> > **Response to authors**
> >
> > I thank the authors for their detailed response to my questions. I encourage them to integrate the explanation for the hyperparameter transfer into their manuscript (possibly in the appendix if space does not permit in the main paper). I am rising my score.

---

> > > ### Author Response · Authors · 2025-11-24
> > >
> > > Thank you for your thoughtful follow-up and for raising your score. We will incorporate your suggestion on clarifying the hyperparameter transfer into the final version of the manuscript.

---

> > > > ### Comment · Reviewer_nQUU · 2025-11-27
> > > >
> > > > I've gone through the recent comments on the other reviewers. I agree with some of their points, e.g. regarding building better representative batches in a long-sequence setting, but disagree that access to data is an issue. In the majority of CL works that I am aware of, its implicitly assumed to be enough data available for training (in some sense, CL arises because there is, in principle, enough data, but it just comes in over time);  so the authors here build on a standard setting.
> > > > Regarding more (multi-modal or text) datasets and corresponding experiments, I think that this, while an understandable request as of lately, requires extensive compute resources, especially when working with LLM models.
> > > >
> > > > Overall, after going through the recent rebuttals, I further raise my score.

---

### Official Review · Reviewer_6HA7 · 2025-10-27

**Soundness:** 4
**Presentation:** 3
**Contribution:** 3
**Rating:** 8
**Confidence:** 5

**Summary:**

Memory-based methods in Continual Learning focus on learning a sequence of tasks with help of a buffer, which store samples from previous tasks and use them when learning the current tasks. Normally, this buffer is limited in the amount of samples they can store due to some constraints of the environment. In this paper, the authors challenge this constraint mainly because storage costs much less than training time. However, as we increase the samples we store, models could face a different challenge, and instead of having forgetting (too much plasticity) this shifts to a high stability and not learning the current task enough. Considering all this, this paper proposes Weight Space Consolidation, which combines a reset strategy to improve plasticity and weight averaging to enhance stability. Experiments on multiple modalities show good performance of the proposed method as they increase the memory budget. Also, multiple ablation experiments help understand why and how the method performs.

**Strengths:**

- The paper is well motivated and written. The authors raise the challenge that, in current systems, memory is not the most important constraint, as GPU time is more costly. Presenting references and numbers, the paper encourages researchers not to focus on minimising memory size.
    - However, there are scenarios where having a small memory is required, such as when resource constraints are imposed or when the privacy of previously learned data is an issue. These are not mentioned or explained in the paper.
- Along with raising the challenge and presenting a new scenario, the authors analyse the limitations of naively increasing memory size and propose a new approach to increase plasticity and enhance stability in this new scenario.
- The experiments and results clearly help to demonstrate what is discussed in the text. Experiments across multiple benchmarks and modalities provide robustness to the results, and ablation experiments help understand the method's limitations and provide further insights into how it works.

**Weaknesses:**

- A common approach to increase the plasticity of memory-based methods is to concatenate buffer samples with current-task samples at the batch level. This is different from what is shown in the paper, where the full batch is sampled from the intersection of the current data and the buffer. As shown in the paper, this last approach suffers from plasticity because it is least likely to sample data from the current task; however, by concatenating at the batch level, there is no plasticity problem.
    - How does this sampling method affect the scenario, results and the proposed method?
    - Concatenating at the batch level can be more costly (in terms of GPU time), but it may mean not keeping a copy of the model in memory, which can increase the batch size.
- Figure 2b is unclear. The orange line makes it impossible to see the blue lines and compare them.

**Questions:**

- Why not compare against better memory-based methods, like for example:
    - Buzzega, Pietro, et al. "Dark experience for general continual learning: a strong, simple baseline." Advances in neural information processing systems 33 (2020): 15920-15930.
        - It has been shown to scale well with the number of tasks.

---

> ### Author Response · Authors · 2025-11-19
>
> We thank the reviewer for the time and effort in reviewing our paper and suggesting good ways to improve our paper. Below, we have responded to your concerns one by one.

---

> ### Author Response · Authors · 2025-11-19
> **[for Reviewer 6HA7] Response to Weaknesses and Questions**
>
> ## Weaknesses
>
> ***
> ### W1. Effect of Batch Sampling
>
> > "How does this sampling method affect the scenario, results and the proposed method?"
>
> Thank you for your comment, the effect of sampling on CL performance has been studied, where at one point, some works claim that sophisticated sampling techniques (e.g., reservoir) boosts CL under constrained memory. However, more recent works that extensively study that random sampling and their more sophisticated counterparts have no significant differences at scale [1]. We believe this trend would persist also in our abundant memory scenarios.
>
> To support this, we share additional results on the CIFAR-100 benchmark using a reservoir sampling regime (i.e., to ensure that memory batch represents each task fairly)
>
>
>
> Method | Memory 20 (4%) | Memory 200 (40%)
> ---|---|---
> Replay | 49.32% (+0.69) | 71.66% (+0.06)
> Ours | 52.49% (+0.33) | 74.63% (+0.14)
> WA | 61.68% (+0.36) | 71.65% (+0.23)
> iCaRL | 50.34% (+0.39) | 72.89% (+0.20)
> DER | 64.21% (+0.26) | 74.83% (+0.19)
> MEMO | 62.36% (+0.37) | 73.40% (-0.31)
> FOSTER | 65.97% (-0.25) | 73.71% (+0.18)
> BiC | 53.86% (+0.21) | 69.27% (+0.12)
>
>
> *Experiments on CIFAR-100 with advanced sampling. The value within the parenthesis indicates the gap between the default random sampling and the reservoir sampling results.*
>
>
>
> The general trend here is that as the memory size becomes larger, the gap becomes smaller, indicating that under abundant memory, the effect of different sampling technique diminishes.
>
> > "... by concatenating at the batch level, there is no plasticity problem."
>
> Thank you for your comment. Indeed, algorithmically ensuring that each batch includes a specific ratio of current data would enhance plasticity, but this would come at the direct cost of stability. Consequently, this approach would suffer from severe catastrophic forgetting of previous tasks. Moreover, batch-level concatenation incurs excessive GPU costs.
>
> In summary, while we agree that the suggested approach could readily improve plasticity, it cannot fully replace our algorithm due to this severe degradation in stability. Additionally, we would like to emphasize that, as mentioned in Dohare et al. (2024) [2], loss of plasticity is a general phenomenon. Our proposed method offers a robust solution for restoring plasticity, independent of the batch design.
>
> ***
> ### W2. Figure 2b
>
> Thank you for your suggestion! We will update the paper accordingly and get back to you.
>
> ***
>
>
>
> ## Questions
> ***
> ### Q1. Other memory-based baselines
> > "Why not compare against better memory-based methods"
>
> Thank you for your suggestion. Indeed, Buzzega et al. (2020) proposed a strong replay-based baseline that combines exemplar memory with logit distillation. However, the baseline is mainly designed for an *online* CL setting where data arrive as a stream and offline retraining is not viable, whereas our setting (e.g., PyCIL benchmark) assumes an offline task-incremental CL scenario. Also, we conducted extensive experiments with SOTA algorithms implemented in PyCIL.
>
> Again, we thank the reviewer for their constructive review, and would like to know if there are any other comments. We will get back to you with a revised version that reflects your feedback.
>
>
> ***
> ### [References]
>
> [1] Mazana et al., Class-incremental learning: survey and performance evaluation on image classification, TPAMI (2022).
>
> [2] Dohare et al., Loss of Plasticity in Deep Continual Learning, Nature (2024).
>
> ***

---

> > ### Comment · Reviewer_6HA7 · 2025-11-24
> >
> > I thank the authors for their response and additional experiments.
> >
> > I agree that sampling strategies boost CL methods only under constrained memory. The new experiments demonstrate that this is indeed the case in your scenario.
> >
> > As concluded in [2], the loss of plasticity is a general phenomenon in CL, but CL methods will always struggle with the stability-plasticity trade-off. I agree with the authors that guaranteeing elements of the new task in the batch can affect stability (increasing forgetting). However, it is necessary as we increase the number of tasks. For example, when we train on 100 tasks, it becomes difficult for the model to incorporate information from new classes. As we increase the number of tasks, the likelihood of sampling new data decreases, especially when we increase memory (as suggested in the paper), and it may even increase training time due to the low gradient signal from the current task (low plasticity). It is important to study the percentage of elements of the current task that are needed in the batch (it does not need to be 50/50) to maintain a good plasticity-stability trade-off.
> >
> > Concerning new baselines. Although DER was proposed for an online scenario, it is a method that works well for offline CL scenarios and scales better with memory size than iCaRL.
> >
> > I will maintain my score.

---

### Official Review · Reviewer_JQyd · 2025-10-30

**Soundness:** 3
**Presentation:** 2
**Contribution:** 3
**Rating:** 6
**Confidence:** 4

**Summary:**

The authors challenge the assumption of traditional CL. Instead, they argue that, instead of the storage, the GPU time is the main bottleneck. They investigated the scenario where memory is sufficient enough to mitigate forgetting, but full retraining from scratch remains the main challenge. As the authors have discovered that models become biased toward prior tasks and struggle to learn new tasks, they propose Weight Space Consolidation, a lightweight method that combines rank-based parameter resets to restore plasticity with weight averaging to enhance stability.

**Strengths:**

+ The paper investigates a new paradigm/scenario that challenges the previous assumptions. Such new thinking is always valuable.
+ The mathematical formulations are rigorous and I discovered zero mistakes there.
+ The new paradigm is not just an "assumption," they have evidence (Section 3) to empirically demonstrate that the new assumption is valid.

**Weaknesses:**

- While the mathematical formulation is rigorous, the theoretical foundation on why this new approach would work is lacking.
- The proposed approach reads like A (rank-based parameter resets) + B (weight averaging). A + B isn't always necessarily bad, but both A and B are the results of prior work, so what's the technical contribution here with this approach?

**Questions:**

See "Weaknesses." What are the technical contributions (s) of this approach, if the A and B components were previously proposed and employed by existing works?

---

> ### Author Response · Authors · 2025-11-19
>
> We thank you for your comments and feedback, especially for agreeing with us on the new paradigm of abundant exemplar memory and its associated issues (e.g., loss of plasticity and reduced forgetting). Below, we have addressed your concerns point by point.

---

> ### Author Response · Authors · 2025-11-19
> **[for Reviewer JQyd] Response to Weaknesses and Questions**
>
> ## Weaknesses
>
> ***
> ### W1. Theoretical Foundation
> > "While the mathematical formulation is rigorous, the theoretical foundation on why this new approach would work is lacking."
>
> Thank you for pointing this. While we have aimed to explicate our method, it is not easy to justify them completely from a theoretical persective. However, the basic idea is straight forward. In the abundant exemplar memory regime, both *plasticity loss* and *forgetting* is an issue. In response, we combine (1) weight re-initialization to restore plasticity and (2) weight averaging to address forgetting.
>
> In the previous literature, numerous studies have shown that weight averaging, especially in-situ averaging like ours, are known to improve generalization [1,3] and address forgetting [2] by encouraging flat minima in the model's loss landscape (lines 299-306, 322-323). Furthermore, our experiment shows that it contributes to improved sparsity (Fig. 11 and lines 1182-1187), which is known to improve the plasticity by making the model rely less on few critical parameters [4,5] (lines 457-460).
>
> **However**, naively combining the two methods does not result in addressing both issues. In the next response, we explained how our algorithm complements the two weight space operations (**We ask the reviewer to see our next response regarding technical contribution**).
>
> ***
> ### W2. Technical Contribution
> > "The proposed approach reads like A (rank-based parameter resets) + B (weight averaging)."
>
>
> (continuing from the previous question) In our paper, we propose a **simple, cost-effective baseline** that combines two techniques with **strong motivations**.
>
> First, we would like to clarify that our method is well motivated: to handle two practical issues observed in the abundant memory regime (both theoretically and empirically studied in Sec 3.2). On top of that, we add a novel constraint, the GPU cost, which is something most existing works overlook.
>
> In consideration of this, we make major adjustments to the weight re-initialization and averaging components. First, we devised an **cost-efficient method** to track dormant parameters with minimal cost (comparative study in Tab. 4 and Lines 437-444). Second, we suggest a **soft re-initialization** method that mixes previous and current weights during reset (comparative study in Tab. 5 and Lines 446-460), for better plasticity, that showed significant gains in performance. Next, regarding weight averaging, we suggest an **in-situ averaging** technique that occurs during training for bettet stability, compared to post-hoc merging-based methods [6,7]. Lastly, we studied how each technique complements each other by encouraging sparsity through averaging (good for re-initialization) and soft resets (good for averaging) (see Tab. 3 and Fig. 11).
>
> In summary, our work does not merely combine algorithms A and B. We first identified specific plasticity and stability challenges in Continual Learning under the sufficient exemplar memory regime. We then strategically applied a cost-effective model merging solution to address them: using it as an initialization to boost plasticity and as in-situ merging to ensure stability. We believe this problem-solution pairing is a distinct contribution beyond existing algorithms.
>
> We thank the reviewer for mentioning this, and we will seek ways to better convey our technical contributions.
>
>
> ***
> ### [References]
>
> [1] Izmailov et al., Averaging weights leads to wider optima and better generalization, UAI (2018).
>
> [2] Cha et al.,  Cpr: classifierprojection regularization for continual learning, ICLR (2021).
>
> [3] Cha et al.,  SWAD: Domain Generalization by Seeking Flat Minima, NeurIPS (2021).
>
> [4] Golkar et al., Continual Learning via Neural Pruning, (2019).
>
> [5] Dohare et al., Loss of Plasticity in Deep Continual Learning, Nature (2024).
>
> [6] Marouf et al., Weighted Ensemble Models Are Strong Continual Learners, ECCV (2024).
>
> [7] Kozal et al., Continual Learning with Weight Interpolation (2024).

---

### Official Review · Reviewer_pTBZ · 2025-10-31

**Soundness:** 2
**Presentation:** 3
**Contribution:** 3
**Rating:** 4
**Confidence:** 3

**Summary:**

Brief Summary: The paper tackles the task of continual learning. The authors suggest gpu compute cost is the main constraint and find the problem is with plasticity where bottleneck is on struggling to learn new tasks. To address this, authors propose Weight Space Consolidation involving rank-based parameter resets and weight averaging. Experiments are conducted on both image-classificaiton tasks (cifar-100) and llm instruction tuning (trace) showing improvement at significantly reduced costs.

**Strengths:**

Pros:

1. The overall point about gpu compute constraints being more than storage is good. Exploration of such middle-ground strategies makes sense to me. The core idea of reseting certain parameters for plasticity also is interesting.

2. In the high-sample regime, the proposed method almost always outperforms existing baselines by 2-3 points.

3. The paper has nice ablation experiments such as comparing replay with reset (table 3), reset strategies (table 5). The plasticity loss experiment (Fig2) in particular suggest the issue with lower plasticity with increasing size of past examples. The additional experiments in the supplementary are appreciated.

**Weaknesses:**

Cons:

1. My main concern is with the framing of the paper for continual learning and its potential application. The point about GPU cost being dominant with storage is fine, but the main problem is that the data itself might not be available in the first place. So storage was never the real issue, it is access to data. For a practical example, assume we have llama-3.2b instruct model but we don't know what data was used in the base to instruct training. Here, the authors are essentially assuming we have access to the underlying data, which is not the case. A few other points:

(i) While storage cost is not a problem, the cost of downloading setting up s3 buckets high throughput speed for gpu access are all real costs. These need to be highlighted.

(ii) If the authors are motivating it based on cost, some experiments on exact cost saved for practical reference should be provided.

This is not to say the original point about GPU vs storage cost isn't correct, but that storage is not the only factor, data access itself is a big one. The entire argument essentially dilutes the author's novelty.

2. It seems a naive full-fine-tuning baseline is missing in Table 2 ? Difference between proposed method and full-fine-tuned on the same set would be very helpful.

3. Authors primarily explore task-based paradigm only, but it seems the method might be more impactful in task-free settings [Ref1].

4. For LLM datasets, authors only consider TRACE dataset, so tough to know the generalizability.. Given that authors are exploring image-based and llm-based separately, it might be worth repeating experiments on multi-modal datasets as well such as on VisCOLL [Ref2].

5. (Minor) Some qualitative visualizations would be very helpful.

6. (Minor) For Trace experiments, the training hyper-parameters are underspecified. What is the fine-tuning method for llama-3.2-1B? Some experiments using LoRA adaptors would also be interesting.

---
[Ref1]: Aljundi, Rahaf, Klaas Kelchtermans, and Tinne Tuytelaars. "Task-free continual learning." In Proceedings of the IEEE/CVF conference on computer vision and pattern recognition, pp. 11254-11263. 2019.

[Ref2]: Jin, Xisen, Junyi Du, Arka Sadhu, Ram Nevatia, and Xiang Ren. "Visually grounded continual learning of compositional phrases." arXiv preprint arXiv:2005.00785 (2020).


---

Overall Rating: 4/10

The main framing of gpu vs storage cost is not really novel. While the authors get 2-3 points improvement in high-sample regime, it is unclear how interesting that is. Authors miss key experiments such as full-fine-tuning baselines, only considering task-based not task-free, and only one trace dataset for llm experiments.

**Questions:**

Q1. Can plasticity be measured as a metric on an eval set? Currently, plasticity loss is provided in the graph.

---

> ### Author Response · Authors · 2025-11-19
>
> We thank you for your time and effort in reviewing our paper. We appreciate the constructive feedbacks and for acknowledging the GPU cost as the main cost bottleneck of continual learning (CL). Below, we responded to your concerns one by one.

---

> ### Author Response · Authors · 2025-11-19
> **[for Reviewer pTBZ] Response to Weakness 1**
>
> ## Weaknesses:
>
> ***
> ### W1. Concern regarding the setting: Data Access is the actual bottleneck.
>
> > "Here, the authors are essentially assuming we have access to the underlying data"
>
>
> We appreciate this critical comment regarding the scope and the challenge of data availability.
>
> We first clarify that our work operates strictly within the exemplar-based CL, which assumes the ability to store a representative subset of past task data (exemplars) in an exemplar memory. **Our contribution is challenging the historical limited memory constraint by modeling the real-world shift where storage is cheap, but GPU compute time is the true bottleneck**. We designed our algorithm to effectively address the resultant loss of plasticity in the sufficient memory regime while minimizing computational overhead.
>
> The scenario you describe, fine-tuning a model like LLaMA-3.2B Instruct without access to its original pretraining data, involves **constraints outside the scope of exemplar-based CL**, particularly regarding the fundamental right to access and store the data. Our adopted setting in experiments with PyCIL benchmark, by contrast, is common and models deployments where data availability for the exemplar buffer is guaranteed (e.g., proprietary or internal data streams).
>
>
> Nevertheless, in under memory-free regimes, our method can be applied. We report the results of the CIFAR-100 experiment (PyCIL) under zero-memory:
>
> Method | Memory 0 (0%)
> ---|---
> Replay | 26.07%
> Ours | 29.83%
> WA | 42.10%
> iCaRL | 29.03%
> DER | 53.37%
> MEMO | 43.98%
> FOSTER | 26.43%
> BiC | 28.29%
>
> *Experiments on CIFAR-100 with Zero-memory*
>
>
> From these experimental results, we conclude the following: While the zero-memory configuration represents an ideal setting for continual learning, our results confirm that it suffers from severe performance degradation. For example, most algorithms achieve 69-74% accuracy in 40% exemplar memory setting (See Table 2 of the manuscript). However, in 0% exemplar memory shown above, accuracy of all algorithms significantly decreases (e.g., 25-53%). This poses a significant limitation in performance to its practical application in real-world services. In conclusion, utilizing a sufficient amount of exemplar memory yields substantial performance gains, reaching a level of feasibility for real-world application.
>
> Additionally, **we explicitly evaluated a closely related, highly relevant scenario: Continual Instruction Tuning on the TRACE benchmark**, where a pretrained LLM (LLaMA-3.2-1B) is adapted on a sequence of downstream tasks. This result (Figure 3, Table 9) demonstrates our method's efficacy in maintaining plasticity and efficiency in a **current, actively researched CL scenario involving foundation models**.
>
>
> Finally, we agree with the reviewer that the setting of severely restricted data availability (e.g., exemplar-free CL or limited access to foundation model training data) is a valuable, albeit distinct, research direction[1]. We will incorporate a discussion of these constraints into our Limitations section, emphasizing that our cost-efficient approach is highly relevant for organizations (e.g., foundation model developers/managers) where exemplar data access is assured.

---

> ### Author Response · Authors · 2025-11-19
> **[for Reviewer pTBZ] Response to Weaknesses W1-(i) and W1-(ii)**
>
> ### W1-(i). The S3 bucket access cost vs. Data storage cost.
>
> > While storage cost is not a problem, the cost of downloading setting up s3 buckets high throughput speed for gpu access are all real costs. These need to be highlighted
>
> Furthermore, we clarify that for a **fixed exemplar memory size** (defined by the percentage of previous task data retained), **all algorithms incur the same non-GPU overheads**, including data access, S3 setup, and storage costs. Therefore, the GPU cost is the sole differentiating factor in our cost-efficiency analysis. As demonstrated in Figure 1, the x-axis precisely captures this GPU bottleneck, measured directly as the total training time, which aligns with common billing metrics for cloud computing services.
>
> ***
>
> ### W1-(ii). The actual cost?
>
> In all of our experiments, the cost of data access and storage is the same across all algorithms using a standard data procedure used in CL. The cost difference lies in the GPU usage cost usually measured with the training time (e.g., AWS pricing criteria). For instance, the cost of running the algorithms in Tab. 2 (w.r.t. to the GPU training time) CIFAR-100 experiments using a V100 GPU on AWS would be:
>
>
> Method | Memory 20 (4%) | Memory 200 (40%) | Memory 400 (80%)
> ---|---|---|---
> Replay (lowest-possible) | $0.66 | $1.63 | $3.16
> Ours | **$0.77** | **$1.79** | **$3.72**
> WA | $2.81 | $6.07 | $9.49
> iCaRL | $3.01 | $5.30 | $9.13
> DER | $3.72 | $7.80 | $11.88
> MEMO | $4.18 | $9.79 | $14.48
> FOSTER | $4.95 | $14.08 | $16.12
> BiC | $6.53 | $13.41 | $16.42
>
> *Training cost of each algorithm on CIFAR-100*
>
> where header indicates the number of exemplars per class (ratio of memory to full data). The costs vary significantly, but in abundant memory regimes, almost all methods fall behind our method in terms of performance while being substantially costly. Note that these results are based on a small dataset (CIFAR-100). For CL scenarios involving massive datasets (i.e., real-world LLM pretraining datasets), the actual cost of each algorithm scales linearly with the size of the dataset.

---

> ### Author Response · Authors · 2025-11-19
> **[for Reviewer pTBZ] Response to Weaknesses 2 - 6**
>
> ### W2. Naive Full-fine-tuning Baseline
> > "naive full-fine-tuning baseline is missing in Table 2"
>
> We agree that comparing with the simplest baseline is important. Respectfully, in most CL works with an exemplar memory, the Replay method (i.e., resume training from previous task model with current task data + previous task data in the memory buffer) is considered the **simplest baseline**. We have reported this result as 'Replay' in the all the tables.
>
> Additionally, we conducted the full-fine-tuning baseline (i.e., train from **scratch** with current task data + previous task data in the memory buffer **without any additional operations**) for our ablation study and it is reported in Tab. 6 of the manuscript. We generally followed experimental setup of the CL literature (e.g., PyCIL benchmark) and we would like to inform that this approach is commonly not considered as a baseline in the CL literature. Additionally, in lines 57-63, and lines 461-473 of the manuscript, we motivate why full-fine-tuning is not suitable in terms of practicality and cost.
>
> ***
> ### W3. Task-free CL
> > Authors primarily explore task-based paradigm only, but it seems the method might be more impactful in task-free settings [Ref1]
>
> We thank the reviewer for the suggestion regarding the task-free setting. While investigating the task-free regime under the abundant memory constraint is a promising direction, we strategically focused our evaluation on the Task-Incremental and Class-Incremental paradigms. These paradigms constitute the standard and most widely adopted CL evaluation protocols across both image classification (PyCIL) and large language models (TRACE), allowing us to first establish a cost-efficient baseline for these foundational scenarios. We concur that extending Weight Space Consolidation to the task-free regime, where the stability-plasticity trade-off becomes more complex, represents a compelling avenue for future work
>
> ***
> ### W4. Expandability to Multi-modal scenarios
>
> > "Given that authors are exploring image-based and llm-based separately, it might be worth repeating experiments on multi-modal datasets as well such as on VisCOLL"
>
> We appreciate the reviewer's suggestions. The reason we tested on Image (classification ; conventional CL benchmarks) and LLM-based Text (relatively newer benchmarks) is because there is an abundant line of works in each domain. Nevertheless, we agree that your suggestion to expand our research towards multi-modal scenarios.
>
> Regarding Multi-modal benchmarks, we believe that *FoMo-in-Flux* [2] is a more representative benchmark. Notably, the trend of merging-based CL methods (i.e., in-situ merging methods work better than post-hoc merging) aligns with our discoveries. Reflecting your suggestion, we will proceed to evaluating on this setting, but please consider that this benchmark is large in scale and costly, and hence the results may be delayed.
>
> However, we strongly believe that, based on the experimental results on PyCIL and TRACE in the manuscript, our method can be successfully applied to multi-modal CL scenarios.
>
> ***
> ### W5. Qualitative Visualization
>
> Thank you! We would love to improve our paper with better visualization. Are there any detailed suggestions you might give us?
>
> ***
> ### W6. Training hyperparameters and the fine-tuning methods (e.g., LoRA) in TRACE
>
> In all of our experiments, we used the default training hyperparameters provided in TRACE. Extending this, we did not use LoRAs as LLM CL methods in TRACE usually assume full fine-tuning without using LoRA adaptors.
>
> Additionally, a recent paper ('LoRA vs Full Fine-tuning: An Illusion of Equivalence,' NeurIPS 2025) reports that the simple application of LoRA to Continual Learning is not significantly beneficial. Therefore, we believe that LoRA-based CL is a research direction that should be distinguished from full fine-tuning. In this light, please note that our study concentrates on the full fine-tuning scenario with an exemplar memory and remain LoRA-based CL as future work.

---

> ### Author Response · Authors · 2025-11-19
> **[for Reviewer pTBZ] Response to Questions and References**
>
> ## Questions:
>
> ***
> ### Q1. Can plasticity be measured as a metric on an eval set? Currently, plasticity loss is provided in the graph.
>
> To the best of our knowledge, the measure of plasticity using eval data is standard [3,4,5]. In these works, *plasticity* is commonly measured as how well the model adapts to the current task on a held-out val/test split (see Fig.2 (a)), rather than on the training data, to avoid conflating plasticity with overfitting or optimization issues. In addition to this, we have also reported the train loss (see Fig.2 (b)) which directly reflects the learning dynamics on the current task: *if the model cannot reduce the loss even on the training data, this indicates a genuine loss of plasticity*, rather than merely a shift in generalization performance.
>
> Showing both eval-based plasticity and train loss (Fig.2 (b)) therefore helps disentangle underfitting due to reduced plasticity from potential overfitting or evaluation artifacts.
>
> ***
> ### [References]
>
> [1] Kirkpatrick et al., Overcoming catastrophic forgetting in neural networks, PNAS (2017).
>
> [2] Roth et al., A Practitioner's Guide to Continual Multimodal Pretraining, NeurIPS (2024).
>
> [3] Zhang et al., Integrating Present and Past in Unsupervised Continual Learning, CoLLAs (2024).
>
> [4] Dohare et al., Loss of Plasticity in Deep Continual Learning, Nature (2024).
>
> [5] Kim et al., Achieving a Better Stability–Plasticity Trade-Off via Auxiliary Networks in Continual Learning, CVPR (2023).
>
> [6] Harmon et al., Mapping Post-Training Forgetting in Language Models at Scale, (2025).

---

> ### Comment · Reviewer_pTBZ · 2025-11-26
> **Appreciate the detailed responses, increased score to 6**
>
> I thank the authors for sharing the detailed responses and additional experimental results which imo greatly solidifies their main contribution.
>
> Comments on each portion
>
> W1: I still think the data access issue should be the bigger concern, but otherwise I am quite happy with the provided tables in particular the costs table (W1-(ii)), I feel it significantly strengthens the author's arguments and would recommend adding it to appendix.
>
> W2: Noted the new results. Appreciate it!
>
> W3/4: I would still encourage the authors to evaluate on multi-modal and task-free CL.
>
> W5: For qualitative results, some visualizations of things that the proposed method is able to identify compared to prior methods would be good, such as higher forgetting in previous works.
>
> Overall, I am updating my score to 6.

---

### Author Response · Authors · 2025-11-19
**General Response for all Reviewers**

# General Response for all Reviewers

***
We sincerely thank all reviewers for their thoughtful feedback, which unanimously recognized the strong motivation to **shift the focus of Continual Learning (CL) from unrealistic memory constraints to GPU computational cost**, and **validated our empirical finding that in this realistic "abundant memory" regime**, the core challenge moves from catastrophic forgetting (stability) to a severe loss of plasticity. This crucial realization, highlighted as a strength by **Reviewers pTBZ** and **6HA7**, naturally motivated **our core technical contribution: Weight Space Consolidation**, a novel and cost-efficient approach that uniquely combines ranking-based parameter resets to restore plasticity (acknowledged as interesting by **Reviewer pTBZ**) and online weight averaging to enhance stability in a simple, integrated, and scalable framework that consistently outperforms strong baselines while maintaining the lowest computational overhead.

Specifically, against the novelty points raised by **Reviewers nQUU** and **pTBZ**, we believe our method provides a clear technical novelty. It functions as **an in-situ, online operation**, a critical distinction from previous post-training merging, which **actively manages the stability-plasticity dynamic during the learning trajectory**. This enables us to establish a **compute-minimal baseline** that is highly effective in abundant-memory, high-cost CL scenarios.

We have carefully addressed all comments and suggestions and will upload a revised paper incorporating these changes promptly.

---

### Author Response · Authors · 2025-11-21
**General Response for all Reviewers (After Revision)**

# General Response for all Reviewers (After Revision)
***
We have completed our revision based on the initial round of author responses and have uploaded the updated manuscript reflecting the reviewers' suggestions. In this revision, our primary goal was to directly address the major weaknesses identified by each reviewer.

For details regarding specific revisions, please refer to our point-by-point responses for the reviewers. To aid your evaluation, all changes in the revised manuscript are highlighted in **purple**.

We sincerely appreciate the reviewers’ thoughtful and constructive feedback. We are fully committed to incorporating any remaining suggestions in the final camera-ready version to further improve the quality and clarity of the work. Thank you.

***
Below, we have listed the updates made in this revision:

- Experiments under Memory-free/Zero-memory: in Appendix lines 1181-1215 and Tab. 10 -- **[Reviewer pTBZ]**
- Unified term for Naive Full-fine-tuning Baseline / Joint training/ From Scratch / Full retraining: unified as Full retraining -- **[Reviewer pTBZ and nQUU]**
- Discussion regarding theoretical foundations of our method: in Appendix lines 998-1008 -- **[Reviewer JQyd]**
- Ablation regarding sampling techniques (e.g., reservoir sampling): in Appendix lines 1238-1293 and Tab. 11 -- **[Reviewer 6HA7]**
- Ablation regarding longer task sequences: in Appendix lines 1295-1303 and Tab. 12 -- **[Reviewer nQUU]**
- Added ratio of memory to full data (%) in Figure 1 -- **[Reviewer nQUU]**
- Minor changes in Figure 2(b) regarding opacity -- **[Reviewer 6HA7]**
- Minor fixes including (1) 'rank-based -> ranking-based' resetting (line 265), (2) clarification of parameter $\alpha_t$ to $\lambda$ (line 177), (3) added explanation regarding $l$ in Eq. 5 (line 287), (4) fixed mixed-use of $Q$ and $q$ -> unified as $Q$.

---

### Comment · Area_Chair_1FfU · 2025-11-24
**Please engage into discussion with authors and fellow reviewers**

Dear reviewers,
The authors have already provided their responses. Do they address your concerns?
Please engage into the discussion with authors and fellow reviewers.
Thanks!
Best,
AC

---

### Meta-Review · Area_Chair_HbEe · 2025-12-23

**Summary:**

All reviewers ultimately concurred that the reframing of the CL problem setting is timely and well motivated, especially for modern large-scale systems where storage is cheap relative to compute.  The method itself (Weight Space Consolidation) has not being questioned: it is simple, efficient, and clearly described. Reviewers also appreciated the empirical evaluation across both image classification benchmarks and continual instruction tuning for LLMs, as well as the careful ablation studies that probe plasticity and stability effects.

The major concern raised by Reviewer pTBZ was related to the assumption of data accessibility and they questioned how realistic this setting is in practice, particularly for foundation models where past training data may be unavailable. Relatedly, while the GPU-cost framing is compelling, reviewers noted that data access and data movement costs are also non-trivial and initially underemphasized.
About the methodology, there have been few concerns about incremental novelty. In particular core components are based on known techniques, and contribution may lie more in their combination than in a fundamentally new algorithm. The theoretical grounding of why and when this combination works was initially viewed as limited, even though the empirical evidence is strong. Some reviewers pointed out missing or unclear baselines (e.g., full retraining) and an excessive focus on task-aware CL.

**Reviewer Concerns:**

The rebuttal successfully addressed a substantial part of the concerns': reviewers were mostly engaged and explicitly mentioned their willingness to raise their scores. Reviewer JQyd was an exception in this respect: their review was very short and not enough articulated. I will downgrade its impact on my final assessment of the submission.

The major concern by Reviewer pTBZ was reasonably addressed by the rebuttal. While the reviewer still thinks data accessibility is a problem in the general case, they are satisfied with the cost tables which strenghten the arguments on the paper rationale.
Other questions about sampling strategies, longer task sequences, and hyperparameter choices were answered with new ablations and extended experiments, reducing ambiguity around the experimental setup.

Concerns regarding novelty persist, as some reviewers still view the method as a principled but incremental combination of existing techniques. Similarly Reviewer pTBZ remained (minorly) concerned about the lack of experiments on multi-modal data and task-free CL.

**Reviewer Scores:**

The two initially negative reviewers pTBZ and nQUU explicitly acknowledged their intention to raise their scores at least to 6. This was based on solid considerations about the satisfactory responses received in the rebuttal, so I am inclined to think that in the end they would have vouched for acceptance.

Reviewer 6HA7 was positive about the work since the beginning and remained such, despite a not entirely satisfactory rebuttal.

Reviewer JQyd assessment is marginally relevant and there was not further interaction with the Authors. My decision takes this aspect into account.

---

### Decision · Program_Chairs · 2026-01-26

Accept (Poster)